

# Global CRM/RM-scaled nutrient gridded dataset GND13

Michio Aoyama[1,2]

1, Research Institute of Global Change, Japan Agency for Marine-Earth Science Technology, Yokosuka, 237-0061, Japan

2, Center for Research in Isotopes and Environmental Dynamics, University of Tsukuba, Tsukuba, 305-8572 Japan

*Correspondence to*: Michio Aoyama (michio.aoyama@ied.tsukuba.ac.jp)

**Abstract.** A global nutrients gridded dataset that might be the basis for studies of more accurate spatial distributions of nutrient in the global ocean was created and named GND13. During 30 cruises, reference materials of nutrients in seawater or their equivalents were used at all stations, and high-precision measurements were made. The precision of the nutrient analyses was better than 0.2%, and comparability between stations was ensured. Data were collected from all of the hydrographic cruises from which nutrient data were available. Analyses were conducted at 243 crossover stations. Data from cruises with high quality control by using RM/CRM of nutrients were used as references to determine correction factors for adjusting nutrient concentrations obtained during other cruises. Dissolved oxygen concentrations obtained with similar protocols were added to the dataset. Finally, a dataset of nitrate, phosphate, silicate, and dissolved oxygen concentrations was created at latitude and longitude intervals of 0.5° and on 136 isobaric surfaces to depths of 6500 meters. This dataset has already been published at doi:10.17596/0000001 (Aoyama, 2017).

## 1 Introduction

Global oceanic biogeochemical cycles are being significantly altered by the direct and indirect impacts of human activities. It is therefore necessary to obtain accurate information about changes and trends of concentrations of inorganic carbon and dissolved inorganic nutrients in both upper and deep ocean waters. For this information to be of practical use, it is critical that results from different laboratories and from geographically similar ocean waters can be reliably compared with complete confidence. A global consensus about nutrient concentrations requires that there be access to accepted certified

reference materials (CRMs), and there must be a requirement or ethos for the use of these CRMs when oceanic nutrient

concentrations are measured and subsequently when they are recorded in global databases, incorporated in climate models,

and ultimately used to quantify changes to the earth system.

The 2007 IPCC Report highlighted the problem inherent in comparing datasets by stating that "Uncertainties in deep

ocean nutrient observations may be responsible for the lack of coherence in the nutrient changes. Sources of inaccuracy

include the limited number of observations and the lack of compatibility between measurements from different laboratories

at different times" (Bindoff et al., 2007). Analyses of nutrient concentrations from crossover stations have shown biases of

up to 10 % for deep water nutrient data during the last three decades (Aoyama et al., 2013; Tanhua et al., 2009). Results of

inter-laboratory comparison studies since 2003 have shown biases of a similar magnitude among some participant

laboratories (Aoyama et al., 2007; 2008; 2010; 2016; 2018). This pattern indicates that analytical problems may be the main

cause of the large discrepancies in reported deep water nutrient concentrations. The reported results imply that these biases

are also present throughout the water column. These comparisons were based on only a small number of specific studies, but

there are many oceanic nutrient datasets reported, published, and stored on international databases with no references to

CRMs at all. Although this situation has improved somewhat since 201x after CRM of nutrients became available, it is still

difficult to ascertain with total confidence any temporal changes in oceanic nutrient concentrations. We can now detect

changes in deep ocean temperature (and hence heat content) (Levitus et al., 2009; 2012; Kouketsu et al., 2009; Rhein et al.,

2013) because of the excellent comparability of temperature measurements over a number of years. Changes to the carbonate

system parameters in the deep ocean have also been reported with comparability ensured by the use of CRMs (e.g.,

Wanninkhof et al., 2010). Similarly, changes in oceanic oxygen concentrations can now be determined (Stendardo and

20 Gruber, 2012).

Reference materials (RMs) and CRMs for nutrients in seawater have been developed for oceanographic use. These

currently include a Danish RM (Eurofins), NRC-Canada CRM (MOOS-3), a new RM developed by Korea (K-RMS), and

one developed by KANSO-Japan. The reference material for nutrients in seawater produced by KANSO has been used in the

inter-laboratory comparison exercise organized by MRI and IOCCP-JAMSTEC since 2003 (Aoyama et al., 2007; 2008;

2010; 2016; 2018). The results of the latest inter-laboratory comparison exercise (IC), "IOCCP-JAMSTEC 2017/18 Inter-

laboratory Calibration Exercise of a Certified Reference Material for Nutrients in Seawater", are now available (Aoyama et

al., 2018). It is clear from the current results (see Figs. 6, 7, and 8 in the 2017/2018 report) that the normalized cumulative

distributions of nitrate and phosphate were better in 2018 than in previous years. The curves were flatter than the normalized

cumulative distributions in previous IC exercises. The implication is that comparability of nitrate and phosphate analyses

among the laboratories gradually improved from 2008 to 2018. This improvement might be a reflection of the fact that the

number of laboratories that use CRM/RMs was increasing during those years. In contrast with the nitrate and phosphate

results, the normalized cumulative distribution for silicate in 2018 was similar to the distributions in previous years. The

implication is that comparability of silicate analyses among the laboratories did not change much from 2008 to 2018. This

difference of comparability between nitrate/phosphate and silicate analyses can be also seen in the results of correction factor

estimation with uncertainty in this study. In particular, correction factor of silicate were more variable and were associated

with greater uncertainty than the correction factors for nitrate and phosphate. Consensus standard deviations of nutrients

concentrations of nitrate, phosphate and silicate were one order of magnitude larger than the homogeneity of the currently

available CRM/RMs and were about double the reported precision of measurements of the individual laboratories. These IC

results therefore showed that use of CRMs should greatly improve the comparability of nutrient data among laboratories

throughout the world by reducing the magnitude of those standard deviations. The current high level of analytical

performance at many participating laboratories indicates that the use of certified reference materials would establish

traceability. The use of CRM/RMs during global cruises in the CLIVAR (Climate and Ocean: Variability, Predictability and

Change), GO-SHIP (Global Ocean Ship-based Hydrographic Investigations Program), and GEOTRACES projects has been

increasing, and the author has been using CRM/RMs during the cruises of the Japan Agency for Marine-Earth Science and

Technology (JAMSTEC) R/V *Mirai* since 2003. It has also been apparent that the uncertainty at crossover stations has been

smaller during cruises when CRM/RMs were used.

This article describes a global gridded dataset produced using CRM/RM-scaled nutrient concentrations based on key

cruises that used CRM/RM. Several previous publications have provided a synthesis results of data collected by several

projects such as the Global Ocean Data Analysis Project (GLODAP), Carbon in Atlantic Ocean (CARINA), and Pacific

Ocean Interior Carbon (PACIFICA) projects (Key et al., 2009; Suzuki et al., 2013; Olsen et al., 2016). This article is an

effort to establish a global nutrients dataset for which comparability and traceability in space and time are explicitly ensured

based on the use of CRM/RMs of nutrients in seawater. Another positive attribute of this work is that uncertainty of

correction factors could be estimated.

**2 Methods and Data**

Data from 30 cruises that used CRM/RM for quality control of nutrient concentrations in seawater were used to obtain

an accurate picture of the spatial distribution of nutrient concentrations in the ocean. The correction factors for those cruises

were set to 1.00 because comparability of nutrient concentrations was ensured (Sato et al., 2010). For oxygen data, the

factors for 30 cruises were assumed to be 1.00 because the high quality control for nutrient analyses on those 30 cruises

suggested that the oxygen analyses were also of high quality.

**2.1 Data collection and quality control**

Nutrient data from the global ocean were collected from various sources and separated into categories from 1 to 7 (Table

1). Thirty cruises were assigned to category 1. Twenty-five of the 30 key cruises were carried out by R/V *Mirai* during 2003–

2013. The author used RM/CRM on those cruises as working standards for nutrient measurements at all stations to ensure

high quality and comparability among the stations and among the cruises. In the Atlantic Ocean, five cruises were also

selected as category 1 because RM were used on two of the five cruises, and good tracking standards with excellent quality

control were used on the other three cruises. Most of the data in category 2 were obtained from the CARINA project dataset.

Most of the data in category 3 were obtained from the PACIFICA project dataset for the period 1991–2008. Many data were

obtained from the World Ocean Circulation Experiment (WOCE) Global Hydrographic Climatology (WGHC) (Gouretski

and Koltermann, 2004) dataset. That dataset includes data from many cruises during the period 1925–1996, and those data

were assigned to category 4. Category 5 was intentionally blank for future use. Some cruises conducted by the Japan

Meteorological Agency (JMA) were not included in the CARINA, PACIFICA and WGHC datasets. They were therefore

included in the dataset for this study and assigned to category 6. Data from about 80 cruises by the JMA and United States

institutes that were not included in the above categories were assigned to category 7.

Figures 1–4 show the locations of all the stations where data were collected. In these figures, stations in category 1 are

marked in dark blue, stations in category 2 are marked in light blue, and stations in categories 3–7 are marked in red. It is

apparent from these figures that the category 1 cruises did not cover the whole ocean, but if category 1 and 2 data are used to

create a global dataset, the spatial coverage increases to almost all of the ocean, and the resultant dataset is a high quality

global nutrient dataset. In southern hemisphere, to cover the some sea areas where categories 1 and 2 were not there,

categories 3-7 were used.

It is important to do quality control before using this historical dataset because it contains questionable data. In the

WOCE dataset and later, there are quality flags (WOCE Hydrographic Programme Office, 1994). Only data associated with

quality flag 2 (i.e., data quality is good) were therefore used in this study. Because the historical data and some of the data

did not have quality flags, a median filter was used to identify questionable data and questionable data were removed from

the dataset before vertical integration, estimation of correction factors and create global gridded data.

**2.2 Crossover analysis**

In general, stations from each cruise within 250 km of 243 points worldwide were selected if there were data from

several stations from at least a cruise in category 1 and at least, respectively a cruise from category 2. A few exceptions were

crossovers in the Pacific sector of the Southern Ocean, where crossovers were selected from category 2 cruises and category

3–7 cruises or among category 3–7 cruises to expand coverage. Figure 5 shows an example of station locations at P03-P14

crossovers at 24.2°N and 179°named CR081E, where there were two category 1 cruises and two category 3–7 cruises.

Supplementary Fig. S1 shows all station locations at the 243 crossovers. Figure 6 shows examples of vertical profiles of

nitrate concentrations, phosphate concentrations, nitrate-to-phosphate concentration ratios, and silicate concentrations at

crossover CR081. Figure 6 also shows climatological nutrient concentrations in the WGHC dataset and WOA05 dataset for

comparison. There were two cruises conducted in 2005 and 2007, which were category 1 cruises, where RMs were used as

working standards at all stations by the author. The results from those cruises were in good agreement with data collected

within a 250-km radius, and the error bounds (i.e., uncertainties) overlapped completely. In contrast, concentrations from two

5  cruises conducted in 1985 and 1993 were relatively scattered (Fig. 6). To estimate correction factors based on 30 key cruises,

vertical integration between depths of 1000 meters and 2000 meters, 1500 meters   and 2500 meters and 2000 meters and

3000 meters for nitrate, phosphate, silicate and oxygen were done at all stations within each of the 243 crossovers. This

integration was done based on the Akima interpolation method (Akima, 1970). When the number of profiles of a cruise

exceeded 3, the standard deviation of the integrated values was calculated as a metric of uncertainty of correction factor.

10  Table 2 shows the valid number of profiles obtained by vertical integration from 1000 meters to 2000 meter depths, from

1500 meter to 2500 meter depths from 2000 meters to 3000 meter depths for nitrate, phosphate, silicate, and oxygen

concentrations at the P03-P14 crossover stations at 24.2°N and 179°E. As expected from the vertical profiles at the

crossovers, the integrated values in units of $\mu mol\ m^{-2}$ for the two cruises in category 1 (e.g., 49MR0505_2_1 and

49MR0706_1_) agreed to within two standard deviations for all four parameters. The standard deviations of two category 2

15  cruises in 1985 and 1993 were relatively large in general, and there were systematic differences that have already been

identified in previous synthesis work. Because there was assurance of comparability of nutrient concentrations among the 30

key cruises, the author set the correction factor for theses cruises to 1.00. Because the measurement uncertainties during

these cruises were less than 0.5% in general, the uncertainty of correction factors were assumed to be 0.00. For oxygen data,

the correction factors for the 30 cruises were assumed to be 1.00 because the high quality control for the nutrient analyses

20  during the 30 cruises might imply that the oxygen measurements were also high quality analyses.

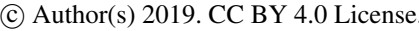

To estimate correction factors at all crossovers the author selected to use integrated values by vertical integration from

1500 meters to 2500 meter depths because smaller Coefficient of Variation (CV), a ratio of the standard deviations of the

integrated values to a mean of the integrated value and largest total number of CV among three integration ranges for nitrate,

phosphate and silicate.

The standard deviation of the integrated values for each of the four parameters can be considered to represent the

combined uncertainty of measurement uncertainty, within-cruise variability (i.e., variability of measurements among several

stations within a 250-km radius), and natural variability among several stations within a 250-km radius at crossovers. It is

interesting to look at the Coefficient of Variation (CV), a ratio of the standard deviations of the integrated values to a mean of

the integrated value of the four parameters (Table 3). Figure 7 also shows histograms of CV of integrated value of the nitrate

data in categories 1–7. It is very clear that the mean of CV of nitrate integrated values were 0.005 for nitrate and phosphate

for category 1 cruises. The corresponding values for silicate and oxygen, 0.009 and 0.018, respectively, were higher than the

values for nitrate and phosphate, although they were smaller than the other values in categories 1–7. The main cause of the

smaller mean of the CV of the integrated values for nutrient concentrations measured during the category 1 cruises was the

use of CRM/RM. The mean of CV of the integrated values for nutrient concentrations were similar to the precision of each

measurement, roughly 0.2–1.0%. The corresponding values for category 1 oxygen measurements were similar to those for

category 2–7 cruises because no reference materials for oxygen measurements exist. Likewise, the silicate measurements

were compromised by some difficulties and/or instabilities—unlike the nitrate/phosphate measurements—that were observed

in the global IC study discussed in the introduction of this article.

During the timeframe of this study, temporal variation of nutrients concentrations within a 250-km radius at crossovers

could be assumed to be of the same and natural variabilities within a 250-km radius in both horizontal and vertical directions

were similar to or smaller than the combined uncertainty of measurement and within-cruise variability (= variability of

measurements among several stations within a 250-radius) based on the data in Table 3. In other words, deep sea water

within a 250-km radius was quite homogeneous horizontally, and the variability of nutrient concentrations in category 2 and

4 cruises might be due to the lower comparability of the nutrient measurements made during those cruises. The larger mean

of the standard deviations of the integrated values for the four parameters at crossovers for the cruises in categories 2–7

might reflect the larger combined uncertainty of measurement uncertainty and within-cruise variability (= variability of

measurements among several stations within a 250-km radius).

When we do factor correction based on synthesis or the method adopted in this study, we need to consider uncertainty of

measurement and within-cruise variability (= variability of measurements among several stations within a 250-km radius)

that might cause the correction factors to be uncertain. The uncertainties of the correction factors were estimated in terms of

the CV of the integrated values within crossovers as a first step in estimation of correction factors. Key cruises included two-

thirds of the 243 crossover points (Figs. 1–4). To estimate correction factors at the remaining crossover points, correction

factor estimations were done progressively. Based on the wider coverage by the cruises in category 2, those cruises were

used as secondary key cruises after correction factors were applied to the integrated values with uncertainty. Factors for

cruises in categories 3–7 were then estimated, with the exception of several crossover stations. Supplementary Table 1 shows

estimated factors and their uncertainties for all cruises.

Comparisons were made between the factors obtained in this study and in GLODAP v2. Figures 8–11 show the results.

For nitrate and phosphate, the correction factors obtained in this study were in good agreement with those obtained by

GLODAP v2 when the correction factors relatively deviated far from 1.00. The implication is that both synthesis work and

direct comparisons as done by this study can detect differences between cruises and estimated correction factors correctly



when the nutrient concentrations obviously differ from values obtained on nearby cruises. For many GLODAP v2 cruises,

factors were assigned a value of 1.00, but it is obvious that direct comparisons resulted in factors that were slightly larger or

smaller (Figs. 7–9) because synthesis work could not identify differences among cruises if those differences were not large.

Direct comparisons, however, could determine correction factors with uncertainties more precisely. In general, the

differences of the correction factors obtained by two methods, synthesis like GLODAP v2 and direct comparison as this

study, for nitrate and phosphate were around +0.02 and –0.04, whereas the differences for silicate were relatively large:

±0.06. For oxygen, the differences were much larger: ±0.10.

**2.3 Gridded dataset**

Based on the factors obtained in this study, a dataset was created at latitude and longitude intervals of 0.5° and on 136

isobaric surfaces at intervals of50 meters. The uncertainties of the nutrient concentrations were about 2% for nitrate and

phosphate and 5% for silicate and oxygen. This uncertainty was equated to twice the standard deviations of the integrated

values for the category 2 cruises. The following steps were used to create the global gridded dataset.

Step 1: Chose profiles of factors determined from the global dataset. Factor corrections and vertical interpolations

were then done for each profile for 136 layers.

Step 2: Before using a surface function of The Generic Mapping Tools, GMT (https://www.soest.hawaii.edu/gmt/),

create overlapping data from –20°E to 0°E and 360°E to 380°E to produce smooth gridded data for the whole

world. Then, conduct a surface function of GMT in each of the 136 layers from the South Pole to 65°N. For the

Arctic Ocean north of 65°N, the latitude and longitude of the data points were converted to an X–Y surface, the



center of which was the North Pole. Then conduct a surface function of GMT for each depth. Convert the

gridded data in the X–Y plane to latitude and longitude at 0.5° intervals.

A gridded dataset with 136 layers at latitude and longitude intervals of 0.5°—the Global Nutrient Dataset 2013— was

then created. Figure 12a–d shows the horizontal distributions of nitrate, phosphate, silicate, and oxygen concentrations at a

5    depth of 1800 meters in the Pacific Ocean as an example.

To determine the total amount of nitrate, phosphate, silicate, and oxygen in the ocean, the volume and area

corresponding to each grid point were calculated using the ETOPO 2 topographic, bathymetric dataset (2-minute mesh). The

concentrations were multiplied by the volume corresponding to the density of the obtained grid point to find the number of

moles of nitrate, phosphate, silicate, or oxygen at each grid point, and the results were summed for each sea area. In this way

10   the total amounts of nitrate, phosphate, silicate, and oxygen in the ocean were estimated as well as the associated

uncertainties.



## 3 Results

This dataset was designated the Global Nutrients Dataset 2013 (GND13). The GND13 is already available at

doi:10.17596/0000001 (Aoyama, 2017) on the JAMSTEC web site. The Fortran source code and ctl script of Grads are also

available at the site. All figures of the horizontal distributions of nitrate, phosphate, silicate, and oxygen are available as

supplementary figures of this article.

Table 4 shows the total mass in petagrams of nitrate, phosphate, silicate, and dissolved oxygen in the ocean. Using the

same methodology, the total petagrams of nitrate, phosphate, silicate, and dissolved oxygen were also calculated for the

WOA09 and WGHC datasets. The total amounts of nitrate, phosphate, silicate, and dissolved oxygen ± uncertainty were 573

± 11 Pg N, 89.0 ± 1.8 Pg P, 3300 ± 170 Pg Si and 7180 ± 360 Pg $O_2$, respectively. As can be seen in Table 4, there were small

differences, but the results were close to the total amounts calculated from the WOA 09 and WGHC climatological

concentrations, which had been published previously and were the initial values of various studies based on a current ocean

general circulation model. Characteristically, nitrate, silicate, and oxygen were estimated to be small, and phosphate was

similar. In addition, the total amount of nitrate was large compared with the literature values: 541 Pg N (Sarmiento and

Gruber, 2006) and 570 Pg N (Wada and Hattori, 1990). The medians of the N:P molar ratios at depths >2 km were 14.6 for

WOA 09 and 14.3 for GND13, and in the latter case the distribution shows high kurtosis (figure not shown). The implication

is that the previous lattice point dataset was generated from a dataset with less comparability, whereas the GND13 dataset

was generated from a dataset with higher comparability.

## 4 Data availability



The GND13 is available at doi:10.17596/0000001 (Aoyama, 2017) at the JAMSTEC web site

http://www.godac.jamstec.go.jp/catalog/data_catalog/metadataDisp/GND13?lang=en.

**5 Conclusions**

A global nutrients gridded dataset, which is the basic dataset used to more accurately characterize the spatial distribution

of nutrients in the global ocean, was created and named GND13. Thirty cruises incorporating reference materials for

nutrients in seawater or their equivalent were used. The precision of the nutrient analyses was better than 0.2%, and

comparability between stations was ensured. Nutrient data were collected from all of the hydrographic cruises from which

nutrient data were available. Crossover analyses were conducted at 243 crossovers where data from our cruises served as

references to determine factors for adjusting nutrient concentrations obtained during other cruises. Dissolved oxygen

concentrations were included as an additional parameter in the dataset using the same protocol. Finally, global datasets of

nitrate, phosphate, silicate, and dissolved oxygen concentrations were created at 0.5° latitude and longitude grid points on

136 isobathymetric layers to a depth of 6.5 km. This dataset will facilitate studies of the behavior of

carbon:nitrogen:phosphorus:oxygen stoichiometry in the ocean in the near future.

**6 Supplement link (will be included by Copernicus)**

xxx

**Author contribution**

Michio Aoyama is the only scientist who created the dataset GND13.

**Competing interests**





The author declares that he has no conflict of interest.

**Acknowledgments**

The author thanks Yukiko Suda and Tomoko Kudo for their work in processing the nutrient data, drawing figures, and

making tables. The author also thanks chemical analysts who measure nutrients concentration which were used in this study,

PIs of nutrients of the cruises and captains and crew of the cruises.

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



Table 1 Summary of data collected and used in this study

| Cruise Category | # of EXPOCODE | # of Profiles | Duration From | To | Main sources |
|---|---|---|---|---|---|
| Nitrate | | | | | |
| 1 | 30 | 2392 | 2003 | 2013 | R/V *Mirai* |
| 2 | 104 | 8857 | 1981 | 2008 | CARINA |
| 3 | 73 | 3598 | 1991 | 2008 | PACFICA |
| 4 | 1235 | 12950 | 1925 | 1996 | WGHC |
| 6 | 7 | 366 | 1996 | 2004 | JMA |
| 7 | 79 | 1931 | 1981 | 2008 | JMA, USA etc. |
| Phosphate | | | | | |
| 1 | 30 | 2392 | 2003 | 2013 | R/V *Mirai* |
| 2 | 102 | 8624 | 1981 | 2008 | CARINA |
| 3 | 72 | 3580 | 1991 | 2008 | PACFICA |
| 4 | 2873 | 30386 | 1925 | 1996 | WGHC |
| 6 | 6 | 345 | 1996 | 2004 | JMA |
| 7 | 78 | 1913 | 1981 | 2008 | JMA, USA etc. |
| Silicate | | | | | |
| 1 | 30 | 2392 | 2003 | 2013 | R/V *Mirai* |
| 2 | 103 | 8850 | 1981 | 2008 | CARINA |
| 3 | 63 | 3207 | 1991 | 2008 | PACFICA |
| 4 | 1870 | 22414 | 1929 | 1996 | WGHC |
| 6 | 6 | 345 | 1996 | 2004 | JMA |
| 7 | 81 | 1862 | 1981 | 2008 | JMA, USA etc. |
| Oxygen | | | | | |
| 1 | 30 | 2319 | 2003 | 2013 | R/V *Mirai* |
| 2 | 109 | 9217 | 1981 | 2008 | CARINA |
| 3 | 77 | 3818 | 1991 | 2008 | PACFICA |
| 4 | 4636 | 49606 | 1906 | 1998 | WGHC |
| 6 | 8 | 426 | 1992 | 2004 | JMA |
| 7 | 73 | 1858 | 1981 | 2008 | JMA, USA etc. |



Table 2 Summary of means and standard deviations of Coefficient of Variation (CV), a ratio of a standard deviations of the integrated values to a mean of the integrated value, within a 250-km radius at crossovers for the 4 variables in each category

| Category | Number of cruises | Mean | Standard deviation |
|---|---|---|---|
| | | Nitrate | |
| 1 | 112 | 0.005 | 0.003 |
| 2 | 381 | 0.012 | 0.013 |
| 3 | 81 | 0.008 | 0.005 |
| 4 | 207 | 0.014 | 0.016 |
| 6 | 14 | 0.008 | 0.005 |
| 7 | 135 | 0.014 | 0.014 |
| | | Phosphate | |
| 1 | 115 | 0.005 | 0.003 |
| 2 | 360 | 0.015 | 0.013 |
| 3 | 80 | 0.009 | 0.006 |
| 4 | 373 | 0.017 | 0.012 |
| 6 | 14 | 0.013 | 0.003 |
| 7 | 132 | 0.016 | 0.014 |
| | | Silicate | |
| 1 | 111 | 0.009 | 0.012 |
| 2 | 352 | 0.030 | 0.032 |
| 3 | 74 | 0.014 | 0.011 |
| 4 | 197 | 0.029 | 0.028 |
| 6 | 12 | 0.016 | 0.010 |
| 7 | 124 | 0.026 | 0.029 |
| | | Dissolved oxygen | |
| 1 | 109 | 0.018 | 0.026 |
| 2 | 390 | 0.014 | 0.029 |
| 3 | 95 | 0.021 | 0.030 |
| 4 | 557 | 0.027 | 0.038 |
| 6 | 13 | 0.019 | 0.017 |
| 7 | 121 | 0.027 | 0.041 |



Table 3 Examples of vertical integration between depths of 1000 m and 2000 m; 1500 m and 2500 m; and 2000 m and 3000 m for nitrate, phosphate, silicate, and oxygen concentrations at CR081, P3-P14 crossovers, 180°E, 24.2°N

| Name | EXPO code | category | 1000-average | 2000 m standard deviation | # of profiles | 1500-average | 2500 m standard deviation | # of profiles | 2000-average | 3000 m standard deviation | # of profiles |
|---|---|---|---|---|---|---|---|---|---|---|---|
| | | | $\mu$mol m$^{-2}$ | | | $\mu$mol m$^{-2}$ | | | $\mu$mol m$^{-2}$ | | |
| | | | | Nitrate | | | | | | | |
| CR081 | 49MR0505_2_1 | 1 | 40874 | 247 | 6 | 39762 | 114 | 6 | 38635 | 71 | 6 |
| CR081 | 49MR0706_1_1 | 1 | 40824 | 404 | 8 | 39685 | 190 | 8 | 38548 | 145 | 8 |
| CR081 | 325023_1_3 | 3 | 40605 | 298 | 9 | 39327 | 82 | 9 | 38174 | 138 | 9 |
| CR081 | 31TTTPS24_2_7 | 7 | 41343 | 215 | 4 | 40119 | 186 | 4 | 38920 | 175 | 4 |
| | | | | Phosphate | | | | | | | |
| CR081 | 49MR0505_2_1 | 1 | 2939.3 | 13.7 | 6 | 2833.4 | 9.7 | 6 | 2735.6 | 6.2 | 6 |
| CR081 | 49MR0706_1_1 | 1 | 2914.9 | 22.3 | 9 | 2809.4 | 8.6 | 9 | 2713.2 | 8 | 9 |
| CR081 | 325023_1_3 | 3 | 2842.4 | 29.4 | 9 | 2730.2 | 20.7 | 9 | 2637.2 | 17.7 | 9 |
| CR081 | 727530_4 | 4 | 3012.9 | | 2 | 2944.9 | | 1 | 2854.7 | | 1 |
| CR081 | 31TTTPS24_2_7 | 7 | 2842.1 | 23.7 | 4 | 2742.4 | 19.1 | 4 | 2649 | 17.7 | 4 |
| | | | | Silicate | | | | | | | |
| CR081 | 49MR0505_2_1 | 1 | 136372 | 2157 | 6 | 150552 | 1112 | 6 | 155601 | 667 | 6 |
| CR081 | 49MR0706_1_1 | 1 | 134493 | 3817 | 9 | 147491 | 2004 | 9 | 151553 | 1080 | 9 |
| CR081 | 325023_1_3 | 3 | 141411 | 4445 | 8 | 154382 | 3018 | 9 | 158403 | 2078 | 9 |
| CR081 | 31TTTPS24_2_7 | 7 | 137563 | 1057 | 3 | 152510 | 1599 | 4 | 157121 | 1043 | 4 |
| | | | | Oxygen | | | | | | | |
| CR081 | 49MR0505_2_1 | 1 | 71030 | 2352 | 6 | 94479 | 1489 | 6 | 113649 | 1211 | 6 |
| CR081 | 49MR0706_1_1 | 1 | 75838 | 8050 | 9 | 96380 | 1875 | 9 | 115548 | 1576 | 9 |
| CR081 | 325023_1_3 | 3 | 73899 | 7080 | 9 | 96608 | 3035 | 9 | 115641 | 1450 | 9 |
| CR081 | 727530_4 | 4 | 82542 | 7715 | 3 | 93094 | | 1 | 109299 | | 1 |
| CR081 | 31TTTPS24_2_7 | 7 | 76793 | 2588 | 4 | 97014 | 1228 | 4 | 116081 | 1561 | 4 |



Table 4 Total amounts of nitrate nitrogen, phosphate phosphorous, silicate silicon, and dissolved oxygen and nitrate vs. phosphate ratios in the global ocean

| | GND13 | WOA09 | WGHC | Sarmiento and Gruber (2006) | Wada and Hattori (1990) |
|---|---|---|---|---|---|
| | Pg | Pg | Pg | Pg | Pg |
| Nitrate nitrogen | 573 ± 11 | 570 | 590 | 541 | 570 |
| Phosphate phosphorus | 89.0 ± 1.8 | 88.3 | 90.5 | | |
| Silicate silicon | 3300 ± 170 | 3330 | 3380 | | |
| Dissolve oxygen | 7180 ± 360 | 7250 | 7240 | | |
| Nitrate vs. phosphate (wt:wt) ratio | 6.44 ± 0.18 | 6.46 | 6.52 | | |
| Nitrate vs. phosphate (mol:mol) ratio | 14.23 ± 0.40 | 14.27 | 14.41 | | |

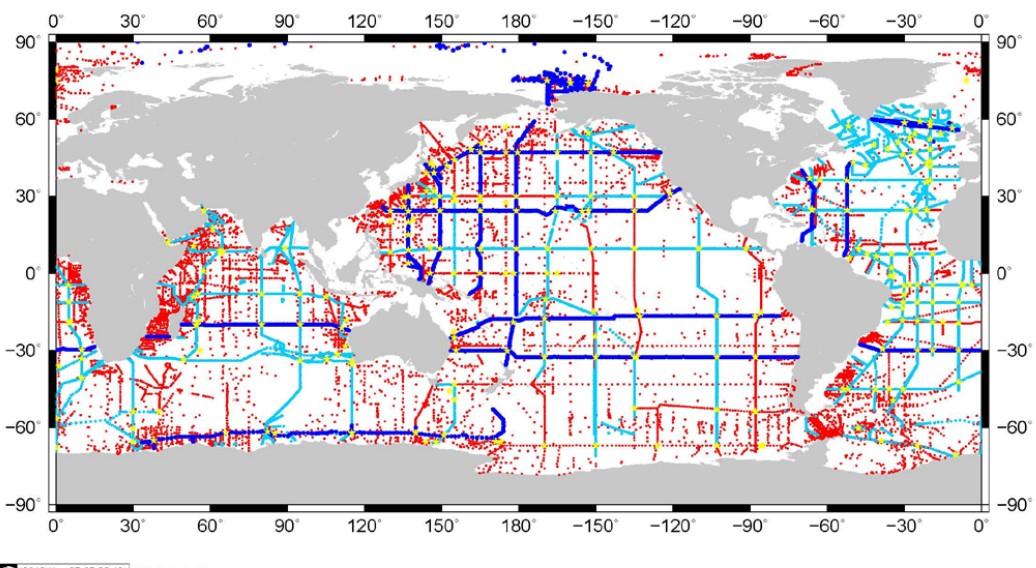

**Figure 1:** Sampling locations where nitrate concentrations were measured and tracks of cruises that measured nitrate concentrations. Dark blue: category 1 cruises with CRM/RM or equivalent quality control. Light blue: category 2 cruises, WOCE/GO-SHIP cruises, but no CRM/RM used. Red: cruises in categories 3–7. Yellow points are crossover points.

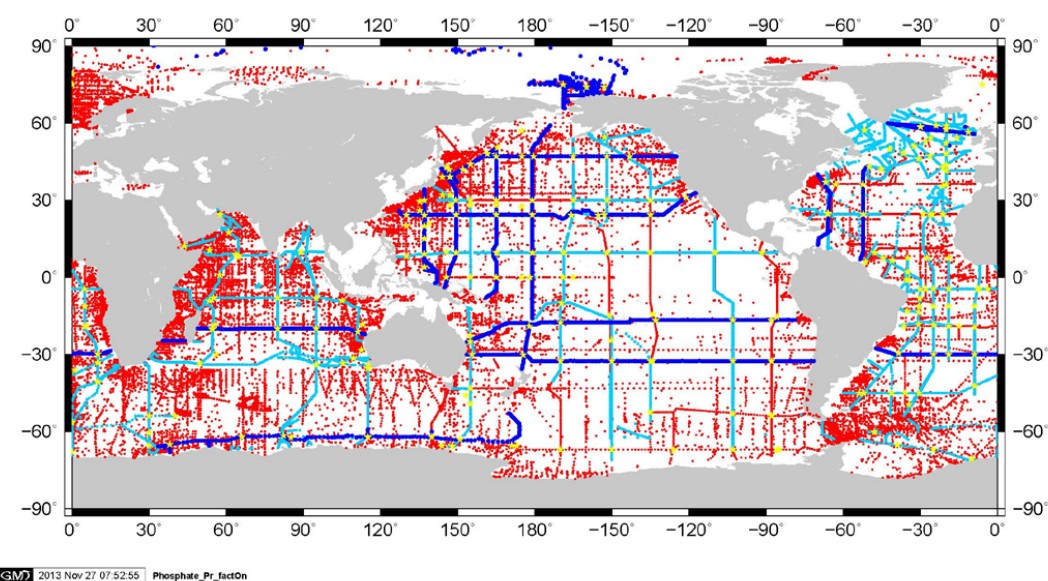

**Figure 2:** Same as Fig. 1, but for phosphate.

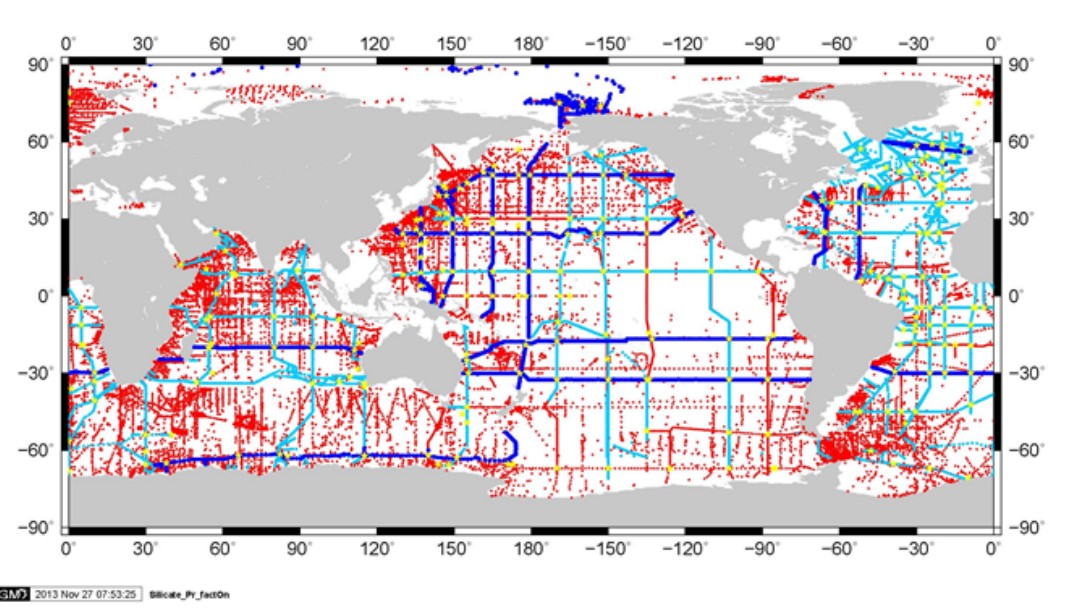

**Figure 3:** Same as Fig. 1, but for silicate.

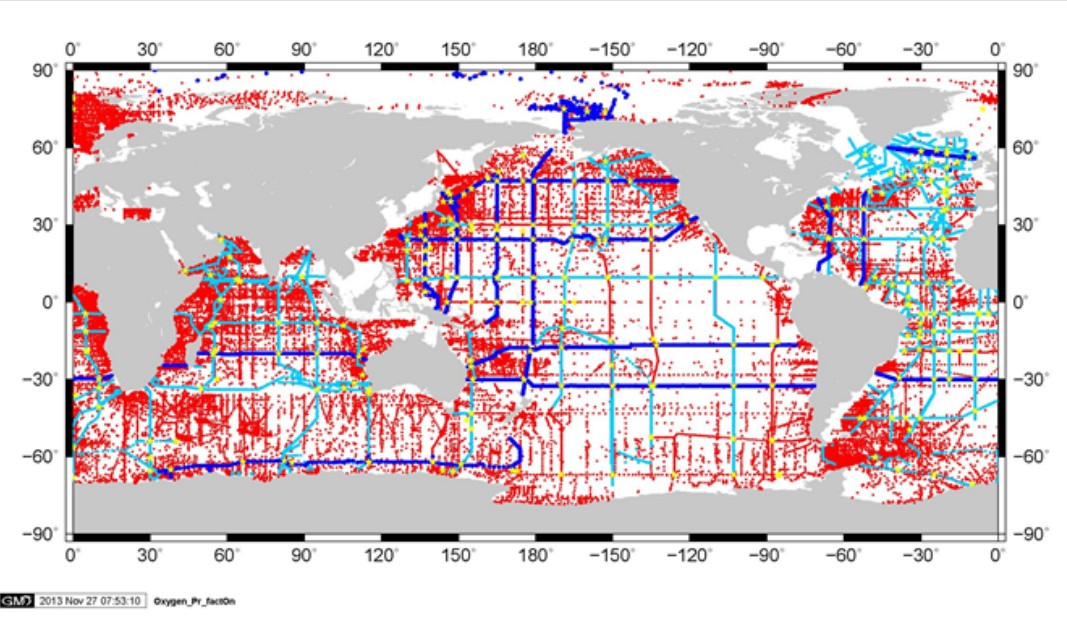

**Figure 4:** Same as Fig. 1, but for oxygen.

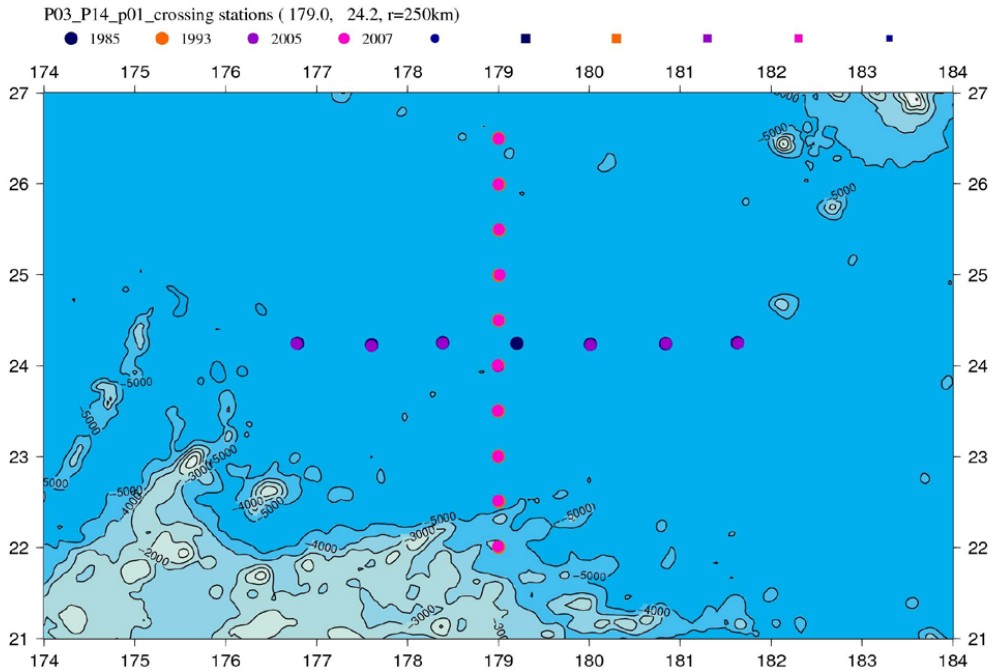

**Figure 5:** Example of crossover stations for comparison at P03-P14 crossover.

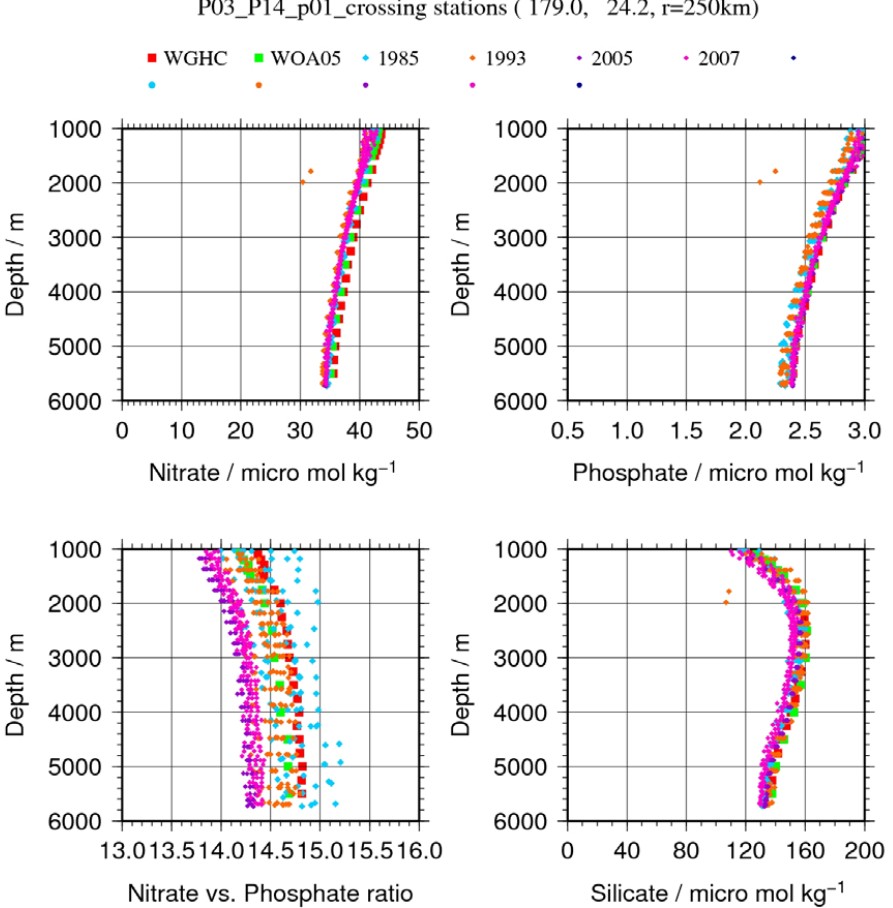

**Figure 6:** Example of vertical profiles of nitrate and phosphate concentrations, the

nitrate:phosphate molar ratio, and silicate concentrations at the P03-P14 crossover (n.b., during the

2005 and 2007 cruises, the author used RM as an in-house standard.)



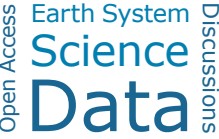

**(a)**

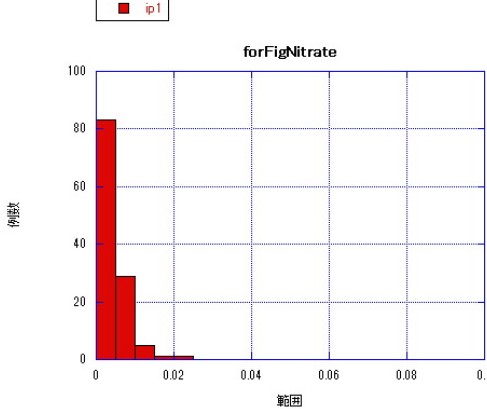

**(b)**

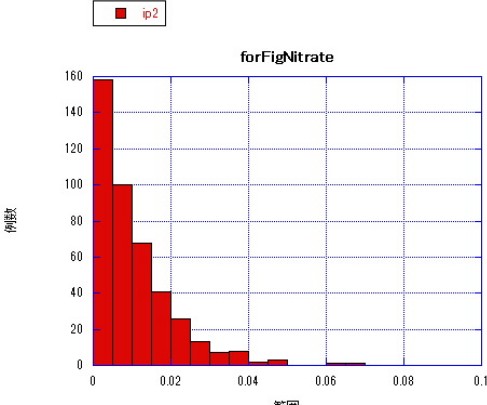

**(c)**

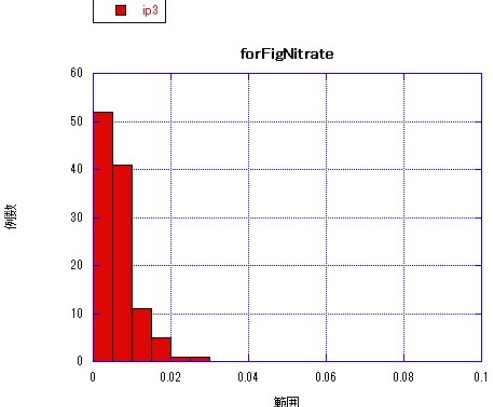





**(d)**

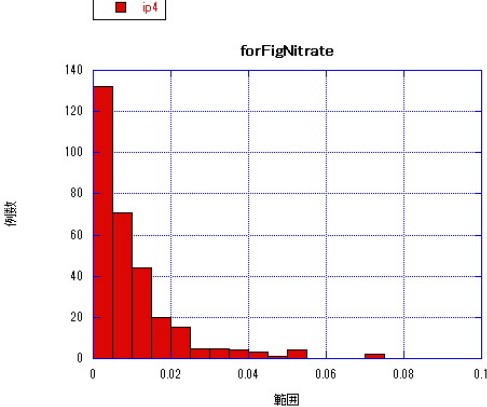

**(e)**

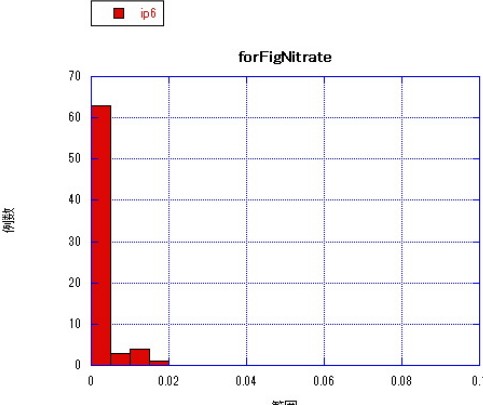

**(f)**

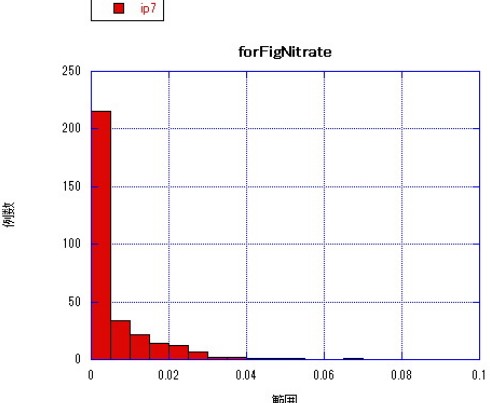



**Figure 7:** Histograms of Coefficient of Variation (CV), a ratio of the standard deviations of the integrated values to a mean of the integrated value, within a 250-km radius for nitrate in (a) category 1, (b) category 2, (c) category 3, (d) category 4, (e) category 6, and (f) category 7.

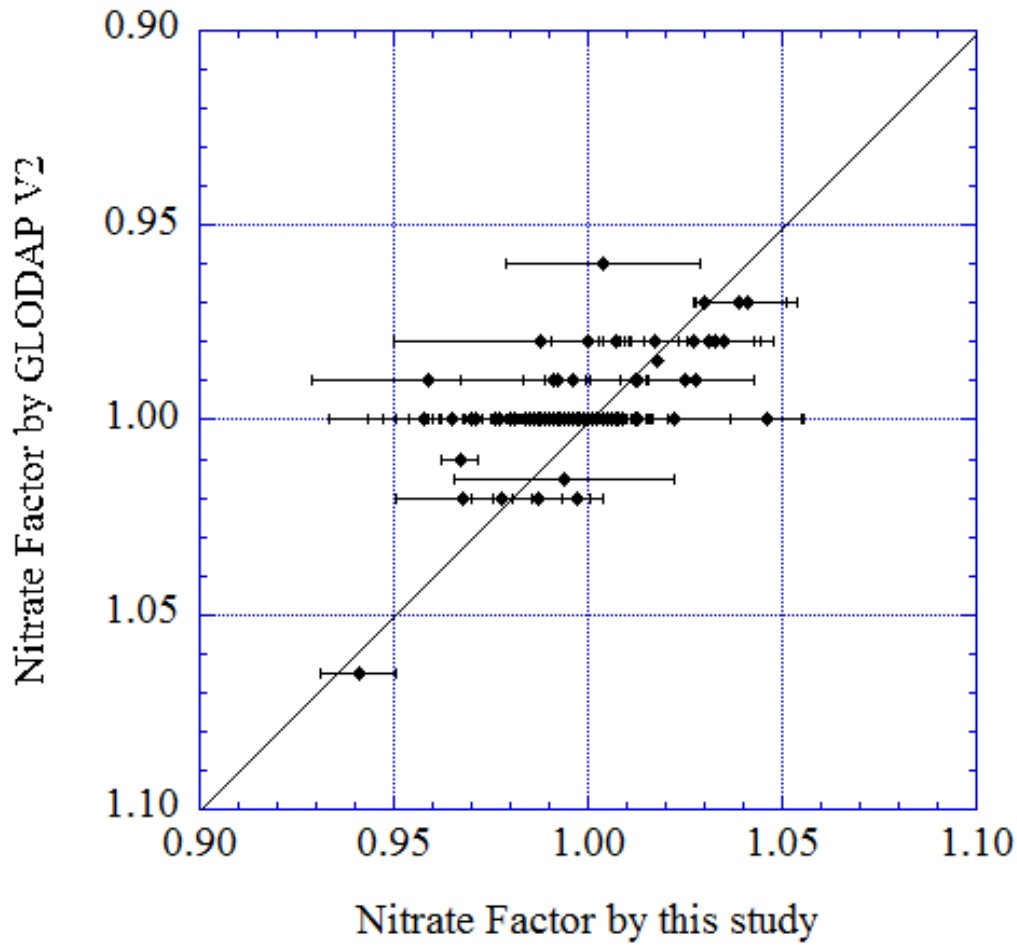

**Figure 8:** A comparison of correction factors for nitrate by this study with uncertainties and by

GLODAP V2.

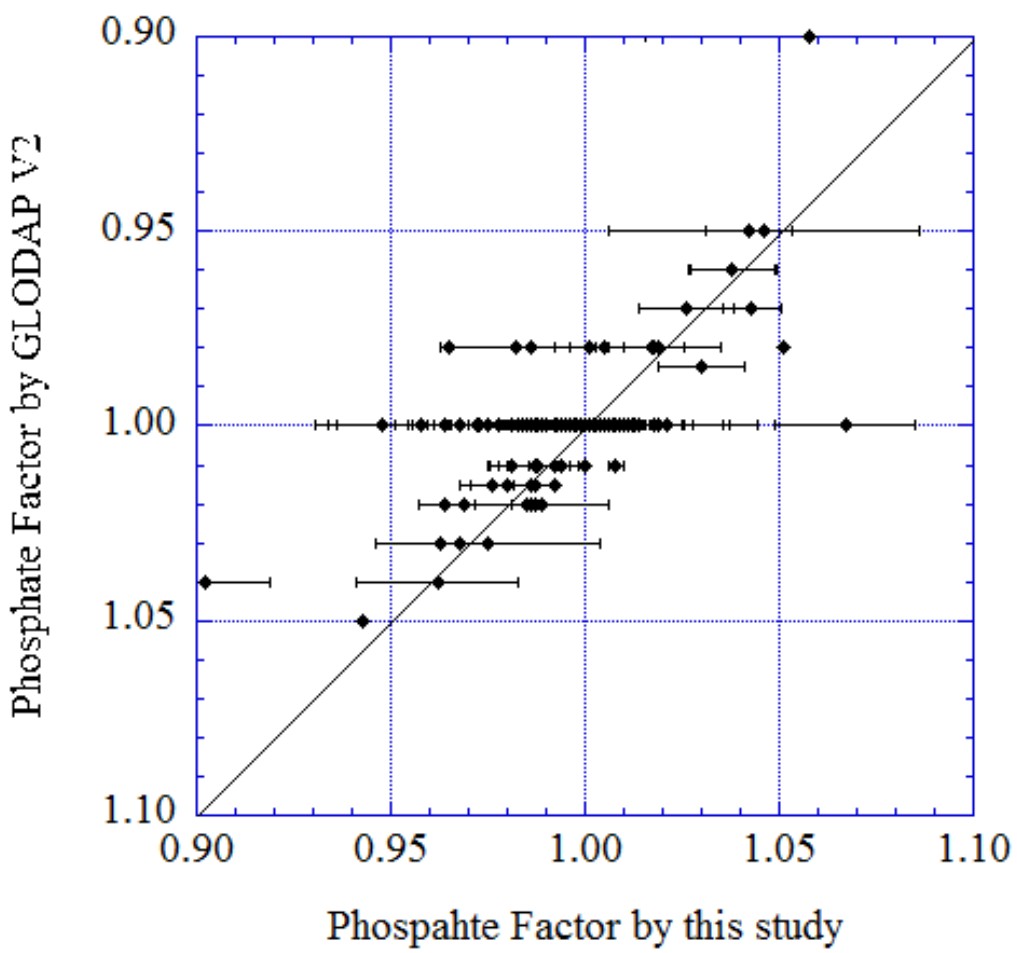

**Figure 9:** Same as Fig. 7, but for phosphate.



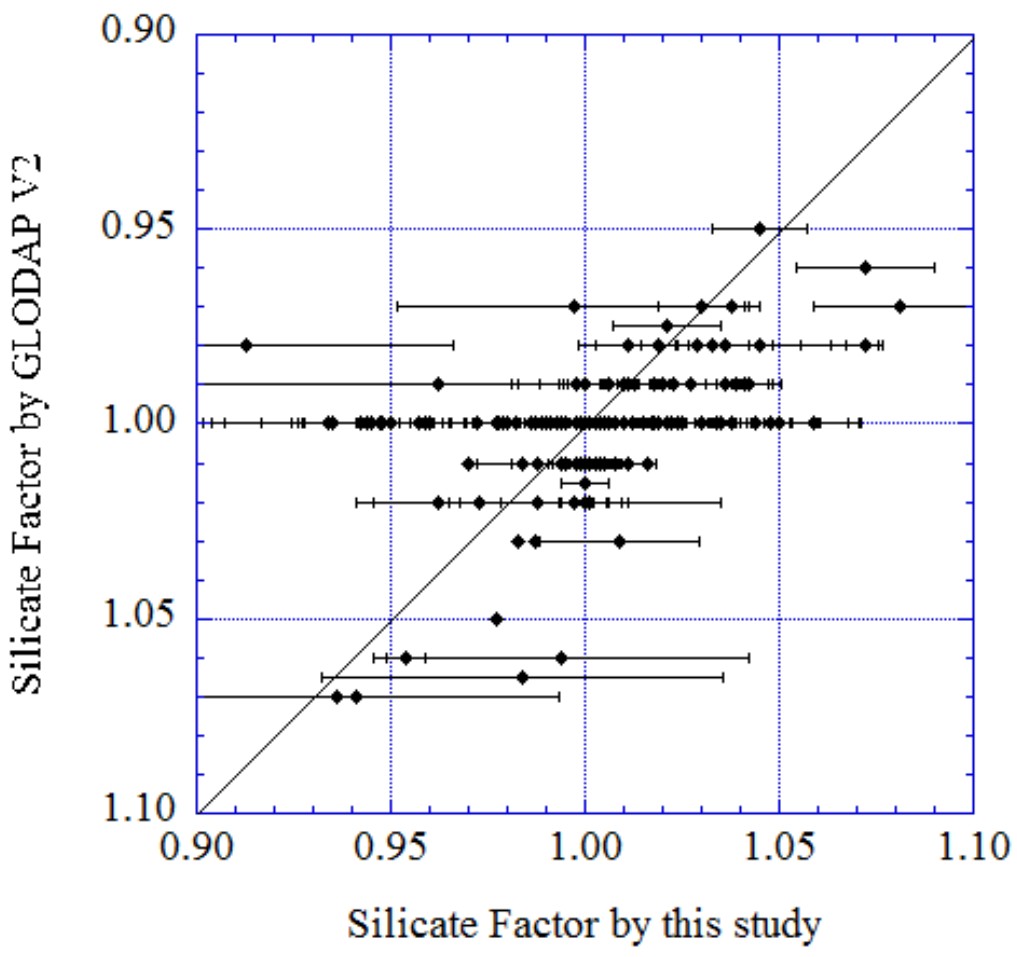

**Figure 10:** Same as Fig. 7, but for silicate.

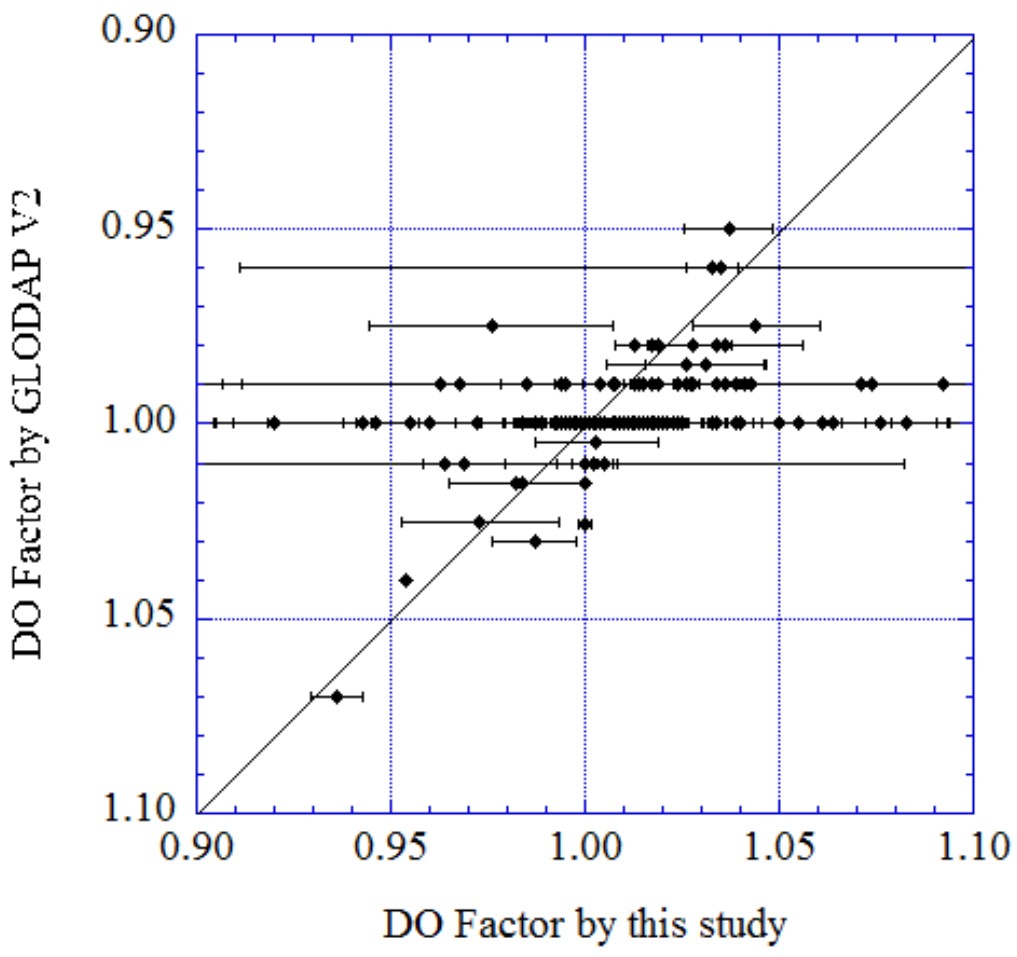

**Figure 11:** Same as Fig. 7, but for dissolved oxygen (DO).





**(a)**

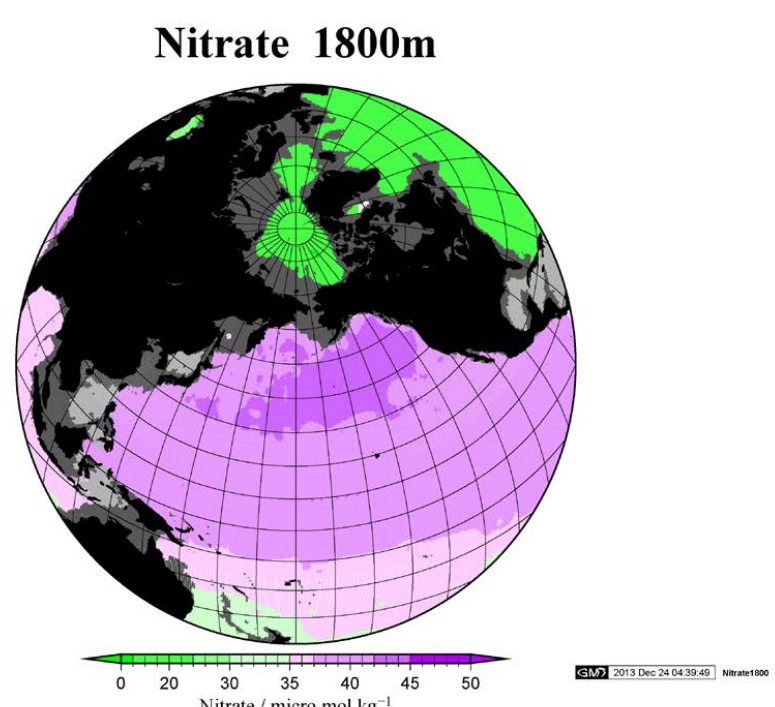

**(b)**





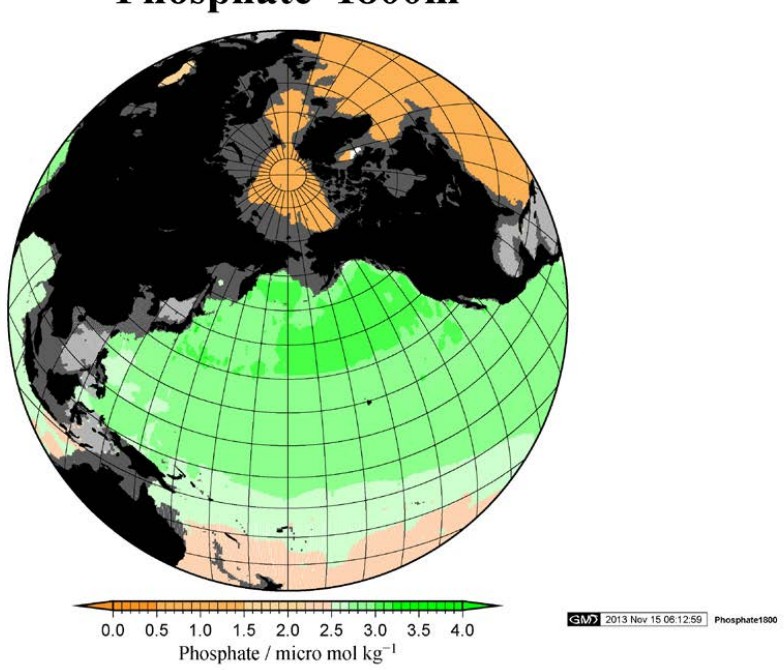

**(c)**

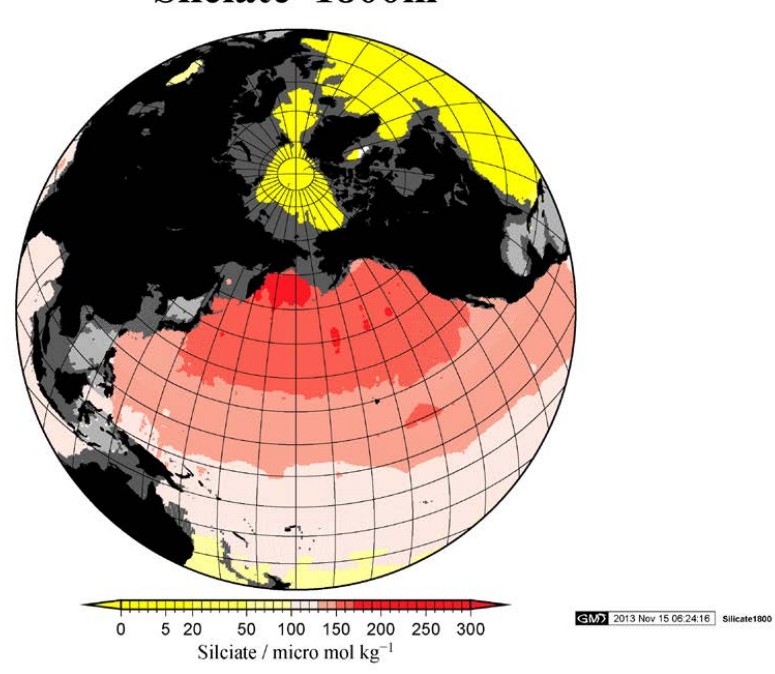

**(d)**

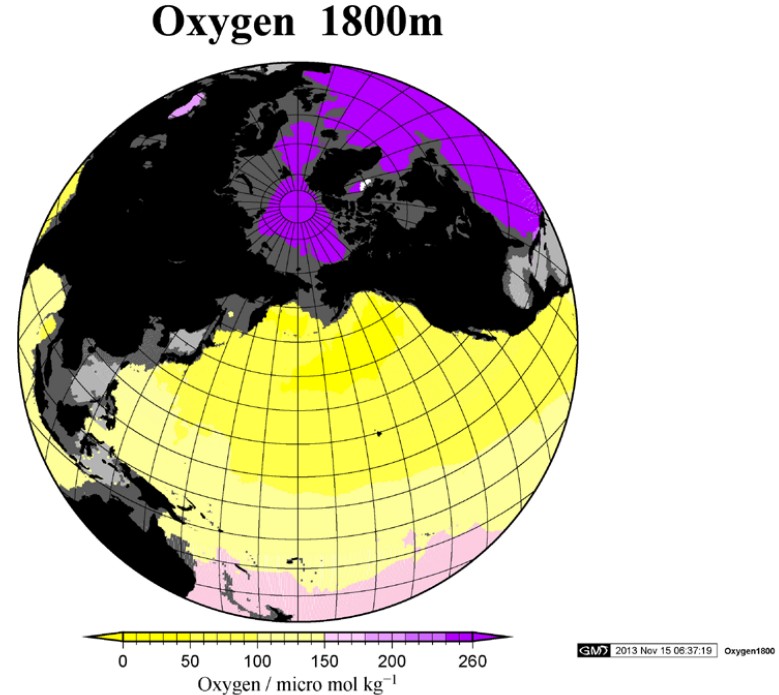

**Figure 12:** (a) Obtained nitrate concentration field at depth of 1800 m, (b) same as (a) but for

phosphate, (c) same as (a) but for silicate, and (d) same as (a) but for dissolved oxygen.