# Peer review of "Global CRM/RM-scaled nutrient gridded dataset GND13"

_Earth System Science Data, 2019_

## Short Comment (SC1) · 23 Aug 2019

Hello Michio,

In your paper you compare data with the World Ocean Atlas 2009. Please use citations for WOA09 as follows:

Oxygen:

Garcia, H. E., R. A. Locarnini, T. P. Boyer, J. I. Antonov, O. K. Baranova, M. M. Zweng, and D. R. Johnson, 2010. World Ocean Atlas 2009, Volume 3: Dissolved Oxygen, Apparent Oxygen Utilization, and Oxygen Saturation. S. Levitus, Ed. NOAA Atlas NESDIS 70, U.S. Government Printing Office, Washington, D.C., 344 pp.

[Figure]

Nutrients:

Garcia, H. E., R. A. Locarnini, T. P. Boyer, J. I. Antonov, M. M. Zweng, O. K. Baranova, and D. R. Johnson, 2010. World Ocean Atlas 2009, Volume 4: Nutrients (phosphate, nitrate, silicate). S. Levitus, Ed. NOAA Atlas NESDIS 71, U.S. Government Printing Office, Washington, D.C., 398 pp.

Thank you,

Hernan

---

## Author Comment (AC1) · 24 Aug 2019

Dear Garcia,

Many thanks for your comments. I will add citations of WOA2009 as you suggested.

Best regards,

Michio

---

## Referee Comment (RC1) · Anonymous Referee #1 · 5 Sep 2019

The author outlines a strategy by which he created a gridded nutrient data product from numerous cruise datasets and data products. He presents some of the metrics used to compare various cruises to one another and then outlines methods he used to turn the merged and adjusted data product into a global gridded data product.

This effort is probably worthwhile, but both ahead of its time and behind its time in some ways.

It is ahead of its time because, unfortunately, nutrient CRMs have not yet been used on enough cruises to afford a global reference data set. There are enough measurements in the North Pacific to justify this exercise, but it is not clear that the same is true in, for example, the Atlantic. Quoting from the paper:

[Figure]

"In the Atlantic Ocean, five cruises were also selected as category 1 because RM were used on two of the five cruises, and good tracking standards with excellent quality control were used on the other three cruises."

It is not clear what "good tracking standards with excellent quality control were used" means, and this seems a weak basis on which to base a data product (or at least a basis that is no better than that used by GLODAPv2). Perhaps this could be reworked to be justified based on deep comparisons between the cruises that did have reference materials and those that category 1 cruises that did not? However, one must select some kind of basis for making a merged and internally consistent data product, so arbitrarily selecting a few cruises and calling them reference lines might also be okay but the language used in the descriptive paper should be more clear that this is what was done.

The paper is behind its time because it follows the release of the GLODAPv2 data product (and its recent 2019 update) which does a similar task and gets similar results with essentially the same data. There are some major differences between GLODAPv2 and this data product: 1. The GLODAPv2 data product update has more and more recent data (this might be a mistaken impression on my part). 2. The GLODAPv2 data product process is more meticulous for all properties excepting perhaps nutrients. The following phrase from the paper suggests very little attention was paid to, for example, oxygen, which is critical co-located data for using nutrient distributions.

"For oxygen data, the factors for 30 cruises were assumed to be 1.00 because the high quality control for nutrient analyses on those 30 cruises suggested that the oxygen analyses were also of high quality."

3. GLODAPv2 has more co-located data types. 4. GLODAPv2 does not make adjustments that are smaller than certain threshold values or adjustments in certain variable regions. This paper suggests this is a flaw, but I would point out that this is done deliberately to avoid erasing any potentially interesting signals in the deep ocean. This nutrient data product assumes no changes at various depths excepting those measured on category 1 cruises. 5. The gridded GLODAPv2 data product presents more detailed gridding methods and better characterizes gridding uncertainties. 6. The nutrient data in this new data product is traceable to CRMs. So, in most respects, the GLODAPv2 product/gridded product and its presentation simply seems to be better, excepting item 6. Item 6 is a very important idea, however, so this ESSD effort could still be very much worthwhile as an exploration of how large of an impact adding traceability would have on nutrient distributions.

Broadly, I think the best thing the author could do would be to work within the GLODAPv2 data product and concentrate the analysis on proposing adjustments to the cruises therein (and any additional cruises newly added) to bring the GLODAPv2 nutrient data in line with CRM-validated sections. This would also be a much more useful exercise for establishing how this process should be done from future data products when more CRM-validated sections are available, and would mean the new data product could benefit from all of the additional co-located data in the GLODAPv2 product.

If the author is not interested in such a significant revision to make the data product more broadly useful, then a much smaller recommendation would be to spend more time and text motivating and justifying the paper. If the author better explains why traceability is critical (which it is for some applications), why oxygen is included in the analysis and why category 1 oxygen cruises were identified in the way they were, how the gridded fields differ from GLODAPv2 gridded fields, etc., then the paper would be nearly publishable. A critical question is how is this data product better than GLODAPv2 (traceability for nutrients, and perhaps there is more data?), since this is what the readers of this paper are going to be wondering. Alternately, if this data product has already been used for several studies and this ESSD paper is just meant to describe how it was created, then a more thorough presentation of what products have used this data product would be useful.

As the paper is currently written, I have a hard time seeing that there would be a large

user-base for this data product... but I could be wrong and there might be many people interested in using it. An advantage of this being a discussion is that they could write in to correct my error if so.

There are also some bits of unclear language to clean up. This should be done via internal review, so I haven't made an exhaustive list of language suggestions.

Line by line comments: Line 10: nutrient → "nutrients" or "nutrient concentrations" Line 12: what is meant by "comparability between stations was ensured" Line 12: collected from which source(s)? Line 14: Suggested rephrasing: Cruises that used certified reference materials (CRMs) for seawater nutrient concentration measurements were used as reference sections to... Line 15: What is meant by similar protocols? Are there O2 reference materials? Line 23: suggestion: "upper and lower" or "shallow and deep" Line 24: suggestion: delete "and from geographically similar ocean waters"... also delete "reliably"... the word reliably is covered by "with complete confidence." Line 25: delete "accepted" since it is redundant with "certified" and the idea, later in the sentence, that people are expected to use the CRMs. P2/L3: "earth" -> "Earth" P2:L9: biases... among -> consistent disagreements... between P2:L15-20: This text seems to imply that deep ocean nutrient changes would be expected if we had more reproducible measurements. However, this paper has not yet presented any literature suggesting that we would expect there to be these changes. I'd recommend adding that literature to the first paragraph of the introduction if any, and being cautious about applying small adjustments based on deep ocean differences over time. P3/L5: ->factors P3/L2 through P3/L6: suggested shortening: The implication is that...among the laboratories did not improve between 2008 to 2018 to the same degree that it did for nitrate/phosphate, and the correction factors for silicate were indeed more variable and uncertain than the correction factors for nitrate and phosphate." P3/L6: nutrients->nutrient P3/L10: suggested delete: "by reducing the magnitude of those standard deviations."... this is unnecessary and it is unclear whether it is referring to the inter-lab deviations or the deviations of the RM homogeneity P3/L15: This is unclear. Perhaps:

[Figure]

Disagreements between cruises at depth tend to be smaller when reference materials are used (then quantify this statement or refer to the section where this information is presented). P3/L18: suggested change to either ->provided a synthesis of... or provided synthesis results of P3/L19: This needs a reference to GLODAPv2. P4/L3: I don't follow the logic... why is the quality of oxygen data high just because nutrient reference materials were used? P4/L11: What is meant by "good tracking standards with excellent quality control"? P4/L12: dataset ->data product P5/L10: What are the median filter parameters? Table 2: if there are 30 cruises in category 1, how are there 112 cruises for the category 1 row for nitrate in table 2? I suspect number of cruises should be number of cruise-intersections or number of profiles used for comparisons. P6/L20: it does not imply that. Also, table 2 perhaps implies that category 2 had more consistently-measured O2 than category 1 despite it coming from a much larger pool of research groups. This suggests the reproducibility of the category 1 oxygen data may be low. P7/L5: vertically integrated? Combined uncertainty of measurement uncertainty? Is this combined measurement uncertainty? Broader question: would it make sense to use density interpolated values or multiple linear regression estimated values to limit the impacts of heaving and shoaling further? P7/L12: This logic doesn't make sense since CRMs were also used for silicate (I think. If I am wrong and they weren't used for silicate then you have the related problem with this logic that the ratio between the category 1 silicate value and the other category silicate values is similar between silicate, nitrate, and phosphate). You explain this later on line 16, but by then the reader is already confused. Just omit the silicate information from line 11. P7/L20 could be assumed to be of the same what? P8/L1: variabilities is defined long after it is used. It is also defined again on lines 6 and 8. P8/L17: GLODAPv2 requires a reference here Fig 7: axis labels are not in English and the figure is low resolution. The figure titles are all the same and confusing. The figure legends are not explained. Category 6 looks very good, yes? Why is this? Fig8: phosphate is miss-spelled. Why are the axes reversed? Since they are reversed, why is a 1:1 line still plotted? P9/L3: It is not that the synthesis could not detect differences when the differences were small, it is that

the people putting the data product together chose not to apply adjustments when the differences were small or potentially real differences. This is an important point, because the approach used by these authors assumes there are no changes in the deep ocean. This means this new data product would eliminate and miss the deep ocean changes that they said motivated their work. P9/L10: Are these gridded uncertainties or uncertainties in the measurements? They seem much too small to account for potential gridding errors. P9/L13: what is meant by "chose profiles of factors determined from the global dataset?" Step 2 is also inadequately explained. P10/L8: multiplied by the volume and the density, yes? It is unclear what is meant by "volume corresponding to the density" P11/L9: how were these uncertainties calculated? P11/L11-12: what is meant by nitrate silicate and oxygen being "small," and phosphate similar? Especially if it is large in the next sentence? P12/L4: suggested deletion: ", which is the basic dataset used to more accurately characterize the spatial distribution of nutrients in the global ocean,"

Side note, there only seems to be 4 cruises in the Atlantic in Fig. 1, is the 5th category 1 cruise the Arctic cruise?

---

## Referee Comment (RC2) · Chris Langdon (Referee) · 6 Nov 2019

The author has made a very substantial contribution by producing a global gridded data set of nutrient data that have been corrected based on RM/CRM to remove systematic biases that were previously obscuring temporal and spatial trends. The work has been thorough and rigorous and the methods clearly documented. This is a data set that will be widely used for many years to come.

One minor item that should be revised appears on Page 7 line 15 where the author says that no reference material of oxygen measurements exists. this is incorrect. OSIL makes a certified potassium iodate standard. These CRMs for oxygen work are not in wide use but should be. With their widespread use it should be possible to reduce

oxygen CVs of cross over comparison from 5% to 1% or better.

---

## Author Comment (AC2) · 4 Dec 2019

1, Many thanks for your detailed review comments. It might took long time to write the comments. I can easily understand and thanks again to take time for this article. Before I reply each comments by referee #1, I would like to state some. We need to recall definition of comparability and traceability in SI, system international. Traceability: property of a measurement result whereby the result can be related to a reference through a documented unbroken chain of calibrations, each contributing to the measurement uncertainty. Metrological comparability: comparability of measurement results, for quantities of a given kind, that are metrologically traceable to the same reference.

[Figure]

The methodology I used in this article is completely follow these definition as shown above and this can ensure comparability and traceability of nutrients part of GND13 dataset with stated uncertainty. I also make clear about comparability and traceability of dissolved oxygen data in this article, namely gridded dataset of dissolved oxygen is NOT SI traceable but traceable to key cruises I selected for nutrients dataset. It is also important that there is also no oceanographic assumption about changes of nutrients in both deep water and shallow waters. I just made unbroken chain of comparison. In terms of time frame, since cruises categorized 1 in this work were conducted between 2003 and 2013, all of the data in this work are for 2003-2013 time frame. In the Pacific Ocean, when we have crossover analyses among category 1 cruises, the comparison showed good consistency within measurement uncertainty. These indicated that during this time frame, nutrients concentration changes in deep water is relatively small compared with uncertainty of measurements. 2, For the dissolved oxygen data, I agree comments of reviewer #1 and add several sentences to make clear the differences between nutrients and dissolved oxygen. Added sentences in page 3 and page 4 about dissolved oxygen comparability and traceability in Introduction are shown below. In page3, I added: On the other hand, the method for determining the dissolved oxygen concentration in seawater is generally the Carpenter method (Carpenter, 1965), which is an improvement of the Winkler method, but is hereafter simply referred to as the Winkler method. In this Winkler method, manganese hydroxide "fixes" dissolved oxygen under alkaline conditions, and the "fixed" dissolved oxygen quantitatively oxidizes iodine ions to free iodine under acidic conditions. Titrating the free iodine with a sodium thiosulfate solution of known concentration indirectly quantifies the dissolved oxygen concentration. The sodium thiosulfate solution concentration is determined by titration of a potassium iodate solution of known concentration (potassium iodate quantitatively oxidizes iodine ions to free iodine under acidic conditions). In Japan, SI-traceable certified reference potassium iodate standards are supplied by the National Meteorology Institute of Japan, National Institute of Advanced Industrial Science and Technology (NMIJ). Ocean Scientific International Ltd, OSIL, UK, and FUJIFILM

Wako Pure Chemical Corporation, Japan, also provides Potassium Iodate solutions, which are used to standardise the thiosulfate solution in the widely used Winkler titration method. Therefore, dissolved oxygen concentration measured around the world had some extent of comparability. In page 4, I added: The author also adds dissolved oxygen concentration data as additional parameter of GND13 using same technology to create nutrients gridded data, unbroken chain of comparison, which means obtained gridded data of dissolved oxygen are traceable to a set of data obtained from 30 key cruises stated in chapter 2 and did not mean SI traceable.

3, Replies to each comments are shown below. Comments: This effort is probably worthwhile, but both ahead of its time and behind its time in some ways. I t is ahead of its time because, unfortunately, nutrient CRMs have not yet been used one enough cruises to afford a global reference data set. There are enough measurements in the North Pacific to justify this exercise, but it is not clear that the same is true in, for example, the Atlantic. Quoting from the paper: "In the Atlantic Ocean, five cruises were also selected as category 1 because RM were used on two of the five cruises, and good tracking standards with excellent quality control were used on the other three cruises." It is not clear what "good tracking standards with excellent quality control were used" means, and this seems a weak basis on which to base a data product (or at least a basis that is no better than that used by GLODAPv2). Perhaps this could be reworked to be justified based on deep comparisons between the cruises that did have reference materials and those that category 1 cruises that did not? However, one must select some kind of basis for making a merged and internally consistent data product, so arbitrarily selecting a few cruises and calling them reference lines might also be okay but the language used in the descriptive paper should be more clear that this is what was done. Reply: Theoretically if only one cruise of category 1 exist in the Atlantic Ocean, after we did "unbroken chain of comparison", comparability and traceability to SI can be established. It is of course uncertainty might be larger due to propagation of error and resulted gridded dataset may not good. In your comment, "It is not clear what "good tracking standards with excellent quality control were used" means." Yes, I agree and I

needed to state more clearly about good comparability about two NIOZ cruises in 2005 and 2007. I add more appropriate explanation in page 5 Line 5 as shown below. in page 5 Line 5: In the Atlantic Ocean, five cruises were also selected as category 1 because RM were used on three of the five cruises. Since comparability of nutrients data between JAMSTEC R/V Mirai cruises during the period from 2003 to 2013 and NIOZ cruises conducted in 2005 and 2007 was explicitly confirmed through inter-laboratory comparison study for reference materials of nutrients in seawater conducted in 2006 and 2008 (Aoyama et al., 2008; 2010), two cruises were added to category 1.

Comments: The paper is behind its time because it follows the release of the GLODAPv2 data product (and its recent 2019 update) which does a similar task and gets similar results with essentially the same data. There are some major differences between GLODAPv2and this data product: I reply to each points you mentioned. 1. The GLODAPv2 data product update has more and more recent data (this might be a mistaken impression on my part). Yes, I agree with you. I used data obtained until 2013. I might use newly collected data and apply my SI traeceable method to new data to update GND13 as GLODAPv2 did.

2. The GLODAPv2 data product process is more meticulous for all properties excepting perhaps nutrients. The following phrase from the paper suggests very little attention was paid to, for example, oxygen, which is critical co-located data for using nutrient distributions. "For oxygen data, the factors for 30 cruises were assumed to be 1.00 because the high quality control for nutrient analyses on those 30 cruises suggested that the oxygen analyses were also of high quality." Yea, as you may understand that salinity and carbonate system parameters have good comparability based on reference materials, eg. IAPSO salinity standard and Dickson's RM, oxygen has potassium iodate solution as reference, and temperature and pressure have SI traceable system. Therefore, major parameters except nutrients have good comparability almost throughout our world and synthesis works like GLODAPv2 can give good results for these parameters. Unlike, based on several Inter-laboratory calibration exercise, comparability of

nutrients should be improved more and theoretically nutrients data should be SI traceable because still many numbers of the cruises were conducted without RM/CRM. For oxygen, I make clear my stance and revised about the oxygen issue stated in another part of my reply and reply to reviewer#2.

3. GLODAPv2 has more co-located data types. Yes, I agree.

4. GLODAPv2 does not make adjustments that are smaller than certain threshold values or adjustments in certain variable regions. This paper suggests this is a flaw, but I would point out that this is done deliberately to avoid erasing any potentially interesting signals in the deep ocean. This nutrient data product assumes no changes at various depths excepting those measured on category 1 cruises. No, I did not say GLODAPv2 has flaw, I stated that synthesis work is a kind of "decision of majority". Therefore, if similar characteristics of chemistry to measure nutrients in some region were dominate and their results were slightly different from SI traceable values and there are no cruises with RM/CRM, synthesis work give their factors are 1.00. But my method, unbroken chain of comparison, says their factors are 1.01, 1.05 etc. as shown in Figures 8-10. I expected this situation before I got my results. These differences between GLODAPv2 and GND13 can be easily understand. I understand data treatment policy in GLODAPv2 that the people putting the data product together chose not to apply adjustments when the differences were small or potentially real differences. But, as I fund and showed figures 8-10, the factors obtained by my work were not small compared with the limit for applying an adjustment in GLODAPv2, eg +-2 % for nutrients as shown in Table 2 in Olsen et al.,(2016). Therefore, since factors I obtained by unbroken chain of comparison based on RM/CRM include larger factors than +- 2% while GLODAPv2 gave factor as 1.00 (Fig. 8-10). These are evidences that synthesis work could not identify differences among cruises if those differences were not large and majority of surrounding regions have similar characteristics while the differences abound a few % might be real because unbroken chain of comparison could detect the differences. Actually based on table S1, it is also noted that for factors assigned as 1.00 by GLO-

DAPv2, 11 of 123 cases were exceed +- 2%, smaller than 0.98 or larger than 1.02, for nitrate and 11 of 107 cases was so for phosphate. Although thses differences are not big, this indicated that differences of methodology made differences about factors estimation. The reviewer also made misunderstanding about an assumption, actually I had/have no assumption, "there are no changes in the deep ocean". One of my interests of nutrients work is to detect nutrients changes in the deep ocean. In this study, as I add new sentences about this issue in Page 8, during the timeframe of this study from 2003 to 2013, temporal variation of nutrients concentrations within a 250-km radius at crossovers at 1500-2500 meter depth was very small and it could be assumed to be negligible based on comparison at crossovers between/among category 1 curies as shown in Figure S1 especially in the Pacific Ocean. I also understand that situation in the Atlantic Ocean may not same and probably temporal variation of nutrients concentrations within a 250-km radius at crossovers at 1500-2500 meter depth was larger rather than that in the Pacific Ocean. Therefore, it is essential to use common CRM to measure nutrients to ensure comparability and traceability of nutrients data in time and space. My work now we are discussing is one of demonstration how CRM works well.

5. The gridded GLODAPv2 data product presents more detailed gridding methods and better characterizes gridding uncertainties. Yes, probably gridding methods used in GLODAPv2 might better rather than simple GMT surface function I used in this work. It is however, I think that important issue is how to create comparable data before girding.

6. The nutrient data in this new data product is traceable to CRMs. So, in most respects, the GLODAPv2product/gridded product and its presentation simply seems to be better, excepting item6. Item 6 is a very important idea, however, so this ESSD effort could still be very much worthwhile as an exploration of how large of an impact adding traceability would have on nutrient distributions. I can show examples of comparison between GLODAPv2 mapping product and GND13 at 2000 m depth along 19.5N and 19.5S as shown below. We look at NP ratio, GLODAPv2 product might a little bit far from reality while GND13 might close to reality due to strong constrain of unbroken

chain of comparison to SI traceable data in category 1(Fig. 1(a) and Fig. 2(a), while we see small differences for nitrate and phosphate concentrations (Figs. 1(b) and (c) and Figs 2(b) and (c)). I also show an comparison of histograms of nitrate vs. phosphate ratio in 130 deg. E – 180 deg. E, 10 deg. N – 30 deg. N and 1500 m – 2500 m region in GLODAPv2 (Fig. 3(a)) and GND13 (Fig. 3(b) as an example. There are clear differences in two histograms, namely GND13 NP ratio showed high kurtosis, sharp peak, compared with GLODAPv2 product NP ratio. These situation was easily expected for me because I observed same situation when I compared GND13 with WOA09 as stated in the main text.

Fig. 1 (a) Nitrate vs. phosphate ratio along 19.5 deg. N at 2000 m in GLODAPv2(red) and GND13(blue).

Fig. 1 (b) Nitrate concentration along 19.5 deg. N at 2000 m in GLODAPv2(red) and GND13(blue).

Fig. 1 (c) Phosphate concentration along 19.5 deg. N at 2000 m in GLODAPv2(red) and GND13(blue).

Fig. 2 (a) Nitrate vs. phosphate ratio along 19.5 deg. S at 2000 m in GLODAPv2(red) and GND13(blue).

Fig. 2 (b) Nitrate concentration along 19.5 deg. S at 2000 m in GLODAPv2(red) and GND13(blue).

Fig. 2 (c) Phosphate concentration along 19.5 deg. S at 2000 m in GLODAPv2(red) and GND13(blue).

Fig. 3 (a) Histogram of nitrate vs. phosphate ratio in 130 deg. E – 180 deg. E, 10 deg. N – 30 deg. N and 1500 m – 2500 m region in GLODAPv2 product.

Fig. 3 (b) Histogram of nitrate vs. phosphate ratio in 130 deg. E – 180 deg. E, 10 deg. N – 30 deg. N and 1500 m – 2500 m region in GND13.

Comments: Broadly, I think the best thing the author could do would be to work within the GLO-DAPv2 data product and concentrate the analysis on proposing adjustments to the cruises therein (and any additional cruises newly added) to bring the GLODAPv2 nutrient data in line with CRM-validated sections. This would also be a much more useful exercise for establishing how this process should be done from future data products when more CRM-validated sections are available, and would mean the new data product could benefit from all of the additional co-located data in the GLODAPv2 product. If the author is not interested in such a significant revision to make the data product more broadly useful, then a much smaller recommendation would be to spend more time and text motivating and justifying the paper.

I partly agree with the reviewer's suggestion and I have a will to collaborate with GLO-DAPv2 team. I am already a good contributor to GLODAPv2 nutrients data through JAMSTEC because I am providing SI traceable nutrients data more than 15 years. I also believe that my GND13 products will contribute to ocean science and contribute to improve comparability of nutrients data because scientists' understanding about big advantage to use CRM when they measure the nutrient concentration will be understood through our discussion and future comparison between GLODAPv2 product and GND13.

Comments: If the author better explains why traceability is critical (which it is for some applications), why oxygen is included in the analysis and why category 1 oxygen cruises were identified in the way they were, how the gridded fields differ from GLODAPv2 gridded fields, etc., then the paper would be nearly publishable. A critical question is how is this data product better than GLO-DAPv2 (traceability for nutrients, and perhaps there is more data?), since this is what the readers of this paper are going to be wondering. Alternately, if this data product has already been used for several studies and this ESSD paper is just meant to describe how it was created, then a more thorough presentation of what products have used this data product would be useful. As the paper is currently written, I have a hard time seeing that there would be a large

user-base for this data product... but I could be wrong and there might be many people interested in using it. An advantage of this being a discussion is that they could write in to correct my error if so.

My reply: In my article, the comparison between GLODAPv2 and GND13 is not an item because this comparison is ongoing work and not only GLODAPv2 but WOA09, WOA13 and WOA18 might include future comparison results. I just show comparison results between GLODAPv2 and GND13 in terms of NP ratio in the reply and not included in the revised articles. But this comparison showed that without traceability we may face small but critical problem on relationship among parameters. Because synthesis work like GLODAPv2 did conduct synthesis for each parameters and did not handle relationship among parameters theoretically some of them have stoichiometric relationship like NP ratio. My method, "unbroken chain of comparison" can keep these stoichiometric relationships, propagated the relationship and made correction to reported nutrients concentration against CRM. This point is advantage and my answer to your critical question "how is this data product better than GLO-DAPv2 (traceability for nutrients, and perhaps there is more data?".

  Comments: There are also some bits of unclear language to clean up. This should be done via internal review, so I haven't made an exhaustive list of language suggestions. Line by line comments: Line 10: nutrient→"nutrients" or "nutrient concentrations" I changed from nutrient to nutrients. Line 12: what is meant by "comparability between stations was ensured" I do not need this sentence, so I deleted this. Line 12: collected from which source(s)? I add sources as "from the hydrographic cruises in JASMTEC R/V Mirai cruises, JMA cruise, CARINA, PACIFICA and WGHC datasets from which nutrient data were available.". Line 14: Suggested rephrasing: Cruises that used certified reference materials (CRMs) for seawater nutrient concentration measurements were used as reference sections to... I changed the sentence based on suggestion and made this statement clear in terms of SI traceability as below; Cruises that used certified reference materials or reference materials (CRMs/RMs) for seawater nutrient

concentration measurements were used as reference of unbroken chain of comparison to determine correction factors which made nutrient concentrations obtained by other cruises to be SI traceable. Line 15: What is meant by similar protocols? Are there O2 reference materials? I changed based on suggestion and made this statement clear in terms of SI traceability as below; Dissolved oxygen concentration data was additional parameter of GND13 using same methodology to create nutrients gridded data, but not traceable to SI. Line 23: suggestion: "upper and lower" or "shallow and deep" I changed as "both shallow and deep ocean waters" Line 24: suggestion: delete "and from geographically similar ocean waters"...also delete "reliably"...the word reliably is covered by "with complete confidence." I did so. Line 25: delete "accepted" since it is redundant with "certified" and the idea, later in the sentence, that people are expected to use the CRMs. I did so. P2/L3: "earth" ->"Earth" I did so. P2:L9: biases...among -> consistent disagreements...between I did so. P2:L15-20: This text seems to imply that deep ocean nutrient changes would be expected if we had more reproducible measurements. However, this paper has not yet presented any literature suggesting that we would expect there to be these changes. I'd recommend adding that literature to the first paragraph of the introduction if any, and being cautious about applying small adjustments based on deep ocean differences over time. Yes, you are correct. I would like to say that deep ocean nutrient changes might be detected if we had more reproducible and SI traceable measurements. I understand your comments very well about deep ocean, I also vary care about deep ocean nutrients changes, too. Unfortunately, there are no literature on this issue yet. Please wait a few years. My colleague and/or I will publish some. P3/L5:->factors I did so. P3/L2 through P3/L6: suggested shortening: The implication is that...among the laboratories did not improve between 2008 to 2018 to the same degree that it did for nitrate/phosphate, and the correction factors for silicate were indeed more variable and uncertain than the correction factors for nitrate and phosphate." I changed these sentences based on your suggestion as "The implication is that comparability of silicate analyses among the laboratories did not improve between 2008 to 2018 to the same degree that it did for nitrate/phosphate,

and the correction factors for silicate were indeed more variable and uncertain than the correction factors for nitrate and phosphate. This improvement might be a reflection of the fact that the number of laboratories that use CRM/RMs was increasing during those years." P3/L6: nutrients->nutrient I did so. P3/L10: suggested delete: "by reducing the magnitude of those standard deviations."...this is unnecessary and it is unclear whether it is referring to the inter-lab deviations or the deviations of the RM homogeneity I did so. P3/L15: This is unclear. Perhaps: Disagreements between cruises at depth tend to be smaller when reference materials are used (then quantify this statement or refer to the section where this information is presented). I followed your suggestion.

P3/L18: suggested change to either ->provided a synthesis of...or provided synthesis results of I changed to "provided synthesis results of" P3/L19: This needs a reference to GLODAPv2. I add a reference in the text and reference list as below; Olsen, A., Lange, N., Key, R. M., Tanhua, T., Álvarez, M., Becker, S., Bittig, H. C., Carter, B. R., Cotrim da Cunha, L., Feely, R. A., van Heuven, S., Hoppema, M., Ishii, M., Jeansson, E., Jones, S. D., Jutterström, S., Karlsen, M. K., Kozyr, A., Lauvset, S. K., Lo Monaco, C., Murata, A., Pérez, F. F., Pfeil, B., Schirnick, C., Steinfeldt, R., Suzuki, T., Telszewski, M., Tilbrook, B., Velo, A., and Wanninkhof, R.: GLODAPv2.2019 – an update of GLODAPv2, Earth Syst. Sci. Data, 11, 1437–1461, https://doi.org/10.5194/essd-11-1437-2019, 2019. P4/L3: I don't follow the logic...why is the quality of oxygen data high just because nutrient reference materials were used? OK, you are right. The logic in the current text is not appropriate. I wanted to use same script and database queries to treat oxygen data as well as nutrients data, therefore I state my thought very short. In fact, I know about good quality control during JAMSTEC R/V Mirai cruses based on potassium iodate standard solution stated in several cruise reports (www.godac.jamstec.go.jp/catalog/doc_catalog/metadataList?lang=ja&tab=category&value=%E5%AD%A6%E8%A1%93%A3%E8%A1%93%A3%E8%A1%93%93%E8%A1%93%93%E8%A1%93%93%E8%A1%93%93%E8%A1%93%93%E8%A1%93%93
), therefore I expect high quality data from these cruises. But this is not explained in the first version of this article. I also made clear that oxygen gridded data is not SI traceable data and changed the statement here as below. In page 4: The author also adds dissolved oxygen concentration data as additional parameter of GND13 using

same technology to create nutrients gridded data, unbroken chain of comparison, which means obtained gridded data of dissolved oxygen are traceable to a set of data obtained from 30 key cruises stated in chapter 2 and did not mean SI traceable. P4/L11: What is meant by "good tracking standards with excellent quality control"? I explained as below and changed a statement here as below; In the Atlantic Ocean, five cruises were also selected as category 1 because RM were used on two of the five cruises. Since comparability of nutrients data between JAMSTEC R/V Mirai cruises during the period from 2003 to 2013 and NIOZ cruises conducted in 2005 and 2007 was explicitly confirmed through inter-laboratory comparison study for reference materials of nutrients in seawater conducted in 2006 and 2008 (Aoyama et al., 2008; 2010) , two cruises were also added to category 1 to increase coverage by category 1 cruises in the Atlantic Ocean. P4/L12: dataset ->data product I did so. P5/L10: What are the median filter parameters? 3 time of standard deviation was criteria for outliers. I add this in the text. Table 2: if there are 30 cruises in category 1, how are there112 cruises for the category 1 row for nitrate in table 2? I suspect number of cruises should be number of cruise-intersections or number of profiles used for comparisons. Yes, "number of crossovers" is correct. I had changed the text in Table 2 from "number of cruise" to "number of crossover point". P6/L20: it does not imply that. Also, table 2 perhaps implies that category 2 had more consistently-measured O2 than category 1 despite it coming from a much larger pool of research groups. This suggests the reproducibility of the category 1 oxygen data maybe low. Yes, I agree with your comment. Reproducibility of the category 1 oxygen data might be slightly low from that of the category 2 cruise in general as 1.4 % vs 1.8 %, but 1.8% is still good number. Comparison of Glodapv2 mapping results and GND13 oxygen gridded data at 2000 m depth along 209.5 deg. E (150.5degW) and 329.5degE (30.5 degW) showed in god agreement shown as below.

I had changed the statement here to make clear my aim and I did as below. For oxygen data, the factors for 30 cruises were assumed to be 1.00 because gridded data of dissolved oxygen are aimed to be traceable to a set of data obtained from 30 key cruises.

P7/L5: vertically integrated? Combined uncertainty of measurement uncertainty? Is this combined measurement uncertainty? Broader question: would it make sense to use density interpolated values or multiple linear regression estimated values to limit the impacts of heaving and shoaling further? I did integration of nutrient concentrations vertically using depth coordinate, then I got number in terms of micro mol m-2 for 1000-2000 meters, 1500-2500 meters and 2000-3000 meters. Examples are shown in Table 3. We have several profiles in each crossover point and standard deviation of integrated values of a set of profiles from each cruise represent the combined uncertainty which should include the uncertainty of measurement, within-cruise variability (i.e., variability of measurements among several stations within a 250-km radius, another word station-station variability) and natural variability among several stations within a 250-km radius at crossovers. I may not understand exact meaning of your broader question above, but I did interpolate by depth coordinate, not density coordinate. And density coordinate interpolation may work well to limit the impacts of heaving and shoaling further based on physical oceanographic knowledge. I changed the sentence to make clear what I want to say here as below. The standard deviation of the integrated values for a set of profiles from each cruise within crossovers can be considered as the combined uncertainty of measurement uncertainty at each profile, station-station variability of measurement within a 250-km radius and natural variability of nutrients concentration among several stations within a 250-km radius at crossovers. It is expected that when RM/CRM were used as working standards to get a calibration curve, station-station variability of measurement within a 250-km radius becomes very small while in-house standard was used, station-station variability of measurement within a 250-km radius may contribute to increase combined uncertainty. Therefore, it is interesting to look at the Coefficient of Variation (CV), a ratio of the standard deviations of the integrated values to a mean of the integrated value of the four parameters (Table 3). P7/L12: This logic doesn't make sense since CRMs were also used for silicate (I think. If I am wrong and they weren't used for silicate then you have the related problem with this logic that the ratio between the category 1 silicate value and the other category silicate values

is similar between silicate, nitrate, and phosphate). You explain this later on line 16, but by then the reader is already confused. Just omit the silicate information from line 11. Yes, current text is not good. I changed the sentence to make clear what I want to say here as below. It is very clear that the mean of CV of integrated values were 0.005 for nitrate and phosphate for category 1 cruises and that for silicate was 0.009. The means of CV of integrated values for nitrate, phosphate and silicate were smaller than those for categories 2–7. The main cause of the smaller mean of the CV of the integrated values for nutrient concentrations measured during the category 1 cruises might be the use of CRM/RM. The mean of CV of the integrated values for nutrient concentrations were similar to the precision of each measurement, roughly 0.2–1.0%. It should be also noted that the silicate measurements were compromised by some difficulties and/or instabilities—unlike the nitrate/phosphate measurements—that were observed in the global IC study discussed in the introduction of this article. On the other hand, the corresponding values for category 1 oxygen measurements were similar to those for category 2–7 cruises because there are no seawater matrix reference materials for dissolved oxygen exist and comparability was kept by potassium iodate solution worldwide as similar magnitude.

P7/L20 could be assumed to be of the same what? and P8/L1: variabilities is defined long after it is used. It is also defined again on lines 6 and 8. Yes, current text is not good. I changed the sentence to make clear what I want to say here as below. During the timeframe of this study from 2003 to 2013, temporal variation of nutrients concentrations within a 250-km radius at crossovers at 1500-2500 meter depth was very small and it could be assumed to be negligible based on comparison at crossovers between/among category 1 curies as shown in Figure S1 especially in the Pacific Ocean. Natural variabilities of nutrients within a 250-km radius at 1500-2500 meter depth were similar to or smaller than the combined uncertainty of uncertainty of measurement and station-station variability of measurement within a 250-km radius which were observed based on the data in Table 3 and other crossover points. In other words, deep sea water within a 250-km radius at 1500-2500 meters was quite homogeneous horizontally, and the variability of nutrient concentrations observed in category 2 and 4 cruises might be due to the lower comparability of the nutrient measurements made during those cruises. P8/L17: GLODAPv2 requires a reference here Fig 7: axis labels are not in English and the figure is low resolution. The figure titles are all the same and confusing. The figure legends are not explained. Category 6 looks very good, yes? Why is this? Fig8: phosphate is miss-spelled. Why are the axes reversed? Since they are reversed, why is a 1:1 line still plotted? Yes, I add a reference of GLODAPv2. I had updated Fig. 7 and put English labels. Category 6 is mostly from JMA and JMA are doing good quality control before they use RM/CRM and comparability among the station within each cruise were relatively good. I have update figure 8-10 as high resolution. I corrected phosphate spell correctly. In GND13, factor is defined as target divided by reference, which means not multiply but should be divided. I know and understand traditional way that factor was used to multiply, but I did a kind of normalization to observed values from category 2-7 cruises by dividing with reference values from category 1 cruises. Therefore, factor was reverse side and the axes reversed. Therefore 1:1 line is also OK. P9/L3: It is not that the synthesis could not detect differences when the differences were small, it is that the people putting the data product together chose not to apply adjustments when the differences were small or potentially real differences. This is an important point, be-cause the approach used by these authors assumes there are no changes in the deep ocean. This means this new data product would eliminate and miss the deep ocean changes that they said motivated their work. I do not agree this reviewer comments. I understand data treatment policy in GLODAPv2 that the people putting the data product together chose not to apply adjustments when the differences were small or potentially real differences. But, as I fund and showed figures 8-10, the factors obtained by my work were not small compared with the limit for applying an adjustment in GLODAPv2, eg +-2 % for nutrients as shown in Table 2 in Olsen et al.,(2016). Therefore, since factors I obtained by unbroken chain of comparison with nutrients data obtained based on RM/CRM include larger factors than +- 2% as shown in Fig. 8-10. These are evidences that synthesis

work could not identify differences among cruises if those differences were not large and majority of surrounding regions have similar characteristics while the differences abound a few % might be real because unbroken chain of comparison could detect the differences. The reviewer also made misunderstanding about an assumption, actually I had/have no assumption, "there are no changes in the deep ocean". One of my interests of nutrients work is to detect nutrients changes in the deep ocean. In this study, as I add new sentences about this issue in Page 8, during the timeframe of this study from 2003 to 2013, temporal variation of nutrients concentrations within a 250-km radius at crossovers at 1500-2500 meter depth was very small and it could be assumed to be negligible based on comparison at crossovers between/among category 1 curies as shown in Figure S1 especially in the Pacific Ocean. I also understand that situation in the Atlantic Ocean may not same and probably temporal variation of nutrients concentrations within a 250-km radius at crossovers at 1500-2500 meter depth was larger rather than that in the Pacific Ocean. Therefore, it is essential to use common CRM to measure nutrients to ensure comparability and traceability of nutrients data in time and space. My work now we are discussing is one of demonstration how CRM works well. P9/L10: Are these gridded uncertainties or uncertainties in the measurements? They seem much too small to account for potential gridding errors. No. This uncertainty was equated to twice the standard deviations of the integrated values for the category 2 cruises as stated in P9L11. P9/L13: what is meant by "chose profiles of factors determined from the global dataset?" Step 2 is also inadequately explained. I have revised the sentence for step 1 as below; Step 1: Profiles of which factor were determined were used to create the global gridded dataset. Then nutrients concentrations were corrected by factor and vertical interpolations were then done for each profile on 136 layers. I have revised the sentence for step 2 to make clear what I did as below;

Step 2: To have smooth gridded data at 0 deg. E (=360 deg. E), data obtained step 1 for 0 deg. E to 20 deg. E were copied to 360 deg. E to 380 deg. E region and data for 340 deg. E to 360 deg. E were copied to -20 deg. E to o deg. E. Then to create grideddata a surface function of The Generic Mapping Tools, GMT

(https://www.soest.hawaii.edu/gmt/), were carried out on each of the 136 layers . North of 65 N, the latitude and longitude of the data points were converted to an X–Y surface. Then conduct a surface function of GMT for each depth. Convert the gridded data in the X–Y plane to latitude and longitude at 0.5 deg. intervals.

P10/L8: multiplied by the volume and the density, yes? It is unclear what is meant by "volume corresponding to the density" I deleted words "corresponding to the density" because these were not necessary here. In the future I intend to make density coordinate dataset, so these words were accidentally remained here. P11/L9: how were these uncertainties calculated? This uncertainty was equated to twice the standard deviations of the integrated values for the category 2 cruises as stated in P9L11.

P11/L11-12: what is meant by nitrate silicate and oxygen being "small," and phosphate similar? Especially if it is large in the next sentence? Current text is not good. I had changed as below. As can be seen in Table 4, the results of GND13 were consistent within uncertainty to the total amounts calculated from the WOA 09 and WGHC climatological concentrations, which had been published previously and were the initial values of various studies based on a current ocean general circulation model. The total amount of nitrate by GND13 was large compared with the literature values: 541 Pg N by Sarmiento and Gruber( 2006 ) and close to 570 Pg N by Wada and Hattori (1990). P12/L4: suggested deletion: ", which is the basic dataset used to more accurately characterize the spatial distribution of nutrients in the global ocean, I deleted as you suggested. "Side note, there only seems to be 4 cruises in the Atlantic in Fig. 1, is the 5th category1 cruise the Arctic cruise? In Fig.1 NIOZ cruises locate very close each other, therefore it may hard to distinguish. End of reply. Note: Figures are in supplement pdf.

Please also note the supplement to this comment:
https://www.earth-syst-sci-data-discuss.net/essd-2019-135/essd-2019-135-AC2-supplement.pdf

---

## Author Comment (AC3) · 4 Dec 2019

Many thanks for your very positive words. I also believe that my GND13 dataset will be widely used for many years to come.

Your comment: One minor item that should be revised appears on Page 7 line 15 where the author says that no reference material of oxygen measurements exists. this is incorrect. OSIL makes a certified potassium iodate standard. These CRMs for oxygen work are not in wide use but should be. With their widespread use it should be possible to reduce oxygen CVs of cross over comparison from 5% to 1% or better. My reply: For the certified potassium iodate standards, I recognize that there are several KIO3 solutions in our world. My meaning in the submitted text was that there are no

seawater matrix dissolved oxygen standard of which oxygen concentration is assigned or certifies. KIO3 solution can not directly give unbroken chain of comparison which give explicit SI traceability. So, I changed a sentences at Page 7 line 15 to make clear current situation about comparability and traceability of dissolved oxygen measurement in the text as below.

Page 7 line 15 The corresponding values for category 1 oxygen measurements were similar to those for category 2–7 cruises because no seawater matrix certified reference materials for oxygen measurements exist but some comparability was ensured by potassium iodate solution of known concentration as stated in the Introduction. I also add some explanation about current status of comparability of dissolved oxygen based on potassium iodate solution in the Introduction as below. On the other hand, the method for determining the dissolved oxygen concentration in seawater is generally the Carpenter method (Carpenter, 1965), which is an improvement of the Winkler method, but is hereafter simply referred to as the Winkler method. In this Winkler method, manganese hydroxide "fixes" dissolved oxygen under alkaline conditions, and the "fixed" dissolved oxygen quantitatively oxidizes iodine ions to free iodine under acidic conditions. Titrating the free iodine with a sodium thiosulfate solution of known concentration indirectly quantifies the dissolved oxygen concentration. The sodium thiosulfate solution concentration is determined by titration of a potassium iodate solution of known concentration (potassium iodate quantitatively oxidizes iodine ions to free iodine under acidic conditions). In Japan, SI-traceable certified reference potassium iodate standards are supplied by the National Meteorology Institute of Japan, National Institute of Advanced Industrial Science and Technology (NMIJ). Ocean Scientific International Ltd, OSIL, UK, and FUJIFILM Wako Pure Chemical Corporation, Japan, also provides Potassium Iodate solutions, which are used to standardise the thiosulfate solution in the widely used Winkler titration method. Therefore, dissolved oxygen concentration measured around the world had some extent of comparability. However, the dissolved oxygen concentration determined by the Winkler method includes the concentration of interfering substances in the seawater sample. End of reply

[Figure]

---

## Author Response (AR2)

The author and the other reviewers have addressed my main concerns with this manuscript, and I believe it is ready for publication after some language edits.

Upon reading the response to reviewers, referee #2's comments, and thinking some more, I now agree that this data product will be of wide use. Even if many people do use GLODAPv2 instead of this data product, this paper still makes an important comment about traceability. Moreover, the added comparisons to GLODAPv2 will help the readers put into context how different this data product is and understand how much GLODAPv2 has to gain through a deeper consideration of traceability.

A: Many thanks for positive evaluation to my revised draft especially in terms of traceability. I also thank reviewer#1 for his/her time to read and check my draft very carefully and make suggestions to improve my draft. I believe that I amended my draft based on you suggestions. I also add a few sentences to make clear this article regarding with your comments.

Reply to specific suggestions and confusions:

P1L15: "used as a reference"
A: I did so.

P1L16: "···by other cruises SI traceable." "data was an additional"
A: I amended as below including next comment.
Cruises that used certified reference materials or reference materials (CRMs/RMs) for seawater nutrient concentration measurements were used as a reference of unbroken chain of comparison to determine correction factors which made nutrient concentrations obtained by other cruises SI traceable. Dissolved oxygen was secondarily-quality controlled using the same methodology as was used to create the nutrients gridded data product, but, lacking a traceable standard, the resulting oxygen data product is not SI traceable.

P1L17: "Dissolved oxygen was secondarily-quality controlled using the same methodology as was used to create the nutrients gridded data product, but, lacking a traceable standard, the resulting oxygen data product is not SI traceable."
A: I amended as below including previous comment.
Cruises that used certified reference materials or reference materials (CRMs/RMs) for seawater nutrient concentration measurements were used as a reference of unbroken chain of comparison to determine correction factors which made nutrient concentrations obtained by

other cruises SI traceable. Dissolved oxygen was secondarily-quality controlled using the same methodology as was used to create the nutrients gridded data product, but, lacking a traceable standard, the resulting oxygen data product is not SI traceable.

P1L19: "as a SI"
A: I did so.

P2L18: "after a CRM for nutrients"
A: I did so.

P3L5: Consider putting the 2nd paragraph at the end of the first.
Yes, I did so.
P3L10: "correction factors for silicate"
A: I did so.

P4L6: "oxygen concentrations"
A: I did so.

P4L7: "world have some"
A: I did so.

P4L9 "and an unbroken"
A: I did so.

P4L16: "data, specifically the unbroken chain of comparison, which"
A: I did so.

P4L17: "cruises identified in section 2. As noted, this does not mean the oxygen product is SI traceable."
A: I did so.

P5L6: "an accurate baseline of"
A: I did so.

P5L7: "set to 1.00, indicating no adjustment is applied, because…"
A: I did so.

P5L9: "aimed at being consistent with the data obtained"
A: I did so.

P5L17: "through an inter-laboratory" or through inter-laboratory comparison studies"
A: I did so.

P5L18: "2010), these two"
A: I did so.

P6L12: "cover the regions not covered by category 1 and 2 cruises, categories" Southern Hemisphere should be capitalized.
A: I did so.

P6L16: "3 times the standard" ⋯ were these calculated for each cruise? If so, indicate.
A: Exact explanation on this issue is as follows.
 3 times the standard deviation was calculated in a selected region of which size was 5 degree of latitude and 5 degree of longitude, and of which layer thickness was 2 times to 6 times of layer thickness at corresponding depth of 136 layers used in this study as standard depths of data product, eg. 10, 20, 30, 40, 50, 75, 100, 125, 150,175, 200, 250, 300,,,, 6450 and 6500 meters. At higher latitude, longitude disntance expanded and it kept 500 km width. This size is consistent with a 250-km radius of crossover points. As an example of region setting, latitude between 20 deg. N and 25 deg. N and longitude between 140 deg. E and 148 deg. E and depth between 950 meter and 1150 meter were set for a region of median filter with a criteria of 3 times the standard deviation of the nutrients data in the region.
Therefore, I add a sentence in the main text as below.
This median filter procedure was done to data sets after the nutrients data was divided into sets of data which was involved in a region of which size was about 500 km x 500 km square and of which thickness was 100 meter to 300 meters in the deep ocean and 10-50 meters in the shallow ocean.

P6L17: "data. Questionable"
A: I did so.

P6L18: "and creation of the global gridded data product."
A: I did so.

P6L21: "243 cruise crossover points"
A: I did so.

P6L22 through P7L2: I don't understand this text and it sounds important.
The original sentence was not correct. The author wanted to state a few exceptions about crossovers constructed only category 3–7 cruises in the in the Pacific sector of the Southern Ocean. This issue is already stated in Page 6 Line 12 as "In southern hemisphere, to cover the some sea areas where categories 1 and 2 were not there, categories 3-7 were used." Therefore, I removed the sentence in Page 6 Line 12 and revised the sentence here as below to make clear the explanation of exception on crossovers.
In the Pacific sector of the Southern Ocean, to cover the regions not covered by category 1 and 2 cruises, categories 3-7 were used to construct crossovers.

P7L2: "As an example, Figure 5 shows the locations of the subset of P03 and P14 stations used for a crossover comparison for a crossover at⋯(Include East/West for longitude). This crossover is assigned the designation CR081E, and there were two category⋯"
A: I did so.

P7L10: "on the 30"
A: I did so.

P8L5: Words are missing here⋯ maybe "because there was a smaller"
A: Yes, I add "there was".

P8L10: "nutrient concentrations"
A: I did so.
P8L12: "measurements with a 250-km radius becomes very small. When in-house standards were used, station-⋯ may contribute meaningfully"
A: I did so.

P8L15: CV is defined twice. It should probably be "the CV" in most places.
A: I have deleted second one and amended as "the CV" in all places.

P8L20: "concentrations was similar"
A: I did so.

P9L4: "oxygen and comparability was kept only with potassium iodate solutions."
A: I did so.

P9L6: "nutrient concentrations"
A: I did so.

P9L8: "category 1 cruises, as"
A: I did so.

P9L10: "250-km radius (based on the data···crossover points)"
A: I made correction following all suggestions stated above for P8L20 to P9L10..

P9L15: "uncertainty from both measurement"
A: I did not follow this suggestion and revised the sentence as below to make clear the discussion.
The larger mean of the standard deviations of the integrated values for the four parameters at crossovers for the cruises in categories 2–7 might reflect the relatively larger uncertainty which was combined of uncertainty of measurement and within-cruise variability (= variability of measurements among several stations within a 250-km radius).

P9L17: "When we apply the method adopted in this study, we need to consider uncertainties of measurements"
A: I did so.

P10L3: What is meant by "with uncertainty" ?
A: This uncertainty is of correction factor, so I revised this sentence as below.
Based on the wider coverage by the cruises in category 2, those cruises were used as secondary key cruises after correction factors and their uncertainties were applied to the integrated values.

P10L19: "of 50"
A: I did so.

P11Step1: "First, nutrient and oxygen profiles were interpolated vertically to 136 levels."
I think we need original first step and also make celar what I did, then I revised this part as

below.

Step 1: Profiles of which factor were determined were selected to create the global gridded dataset. Then nutrients and oxygen concentrations were corrected by the factors.

Step 2: Nutrient and oxygen profiles were interpolated vertically to 136 levels.

And original Step2 put Step3

P11L8: "deg. E. Then the gridded data surface function of the… GMT was used to map these interpolated data horizontally for each of the 136 layers." These two steps both need to be rephrased.

A: I did so.

Table 4: "Dissolved oxygen

A: I had corrected this ypo-mistake.

Figure 5: What are the other squares at the top of the figure? What are the Lat/Lon units (degrees N/S, E/W)

A: I deleted several squares those should not be in the figure.

I amended labels of X- and Y- as you mentioned.

Figure 6: use ½ as many axis labels for the lower left panel.

Honest saying, I cannot understand exact meaning of this comment, but I guess that I need to deleted "13.5, 14.5,,," to reduce number of labels of x-axis and I did so.

End of reply.

**Global CRM/RM-scaled nutrient gridded dataset GND13**

Michio Aoyama

5 Research Institute of Global Change, Japan Agency for Marine-Earth Science and Technology and CRiED, University of Tsukuba, Japan

*Correspondence to*: Michio Aoyama (michio.aoyama@ied.tsukuba.ac.jp)

**Abstract.** A global nutrients gridded dataset that might be the basis for studies of more accurate spatial distributions of

10 nutrients in the global ocean was created and named GND13. During 30 cruises, reference materials of nutrients in seawater

or their equivalents were used at all stations, and high-precision measurements were made. The precision of the nutrient

analyses was better than 0.2%. Data were collected from the hydrographic cruises in JASMTEC R/V *Mirai* cruises, JMA

cruise, CARINA, PACIFICA and WGHC datasets from which nutrient data were available. Analyses were conducted at 243

crossover stations. Cruises that used certified reference materials or reference materials (CRMs/RMs) for seawater nutrient

15 concentration measurements were used as a reference of unbroken chain of comparison to determine correction factors

which made nutrient concentrations obtained by other cruises SI traceable. Dissolved oxygen was secondarily-quality

controlled using the same methodology as was used to create the nutrients gridded data product, but, lacking a traceable

[revised manuscript text omitted]
data which was involved in a region of which size was about 500 km x 500 km square and of which thickness was 100 meter
to 300 meters in the deep ocean and 10-50 meters in the shallow ocean. Questionable data were removed from the dataset
before vertical integration, estimation of correction factors and creation of the global gridded data product.

2.2 Crossover analysis

In general, stations from each cruise within 250 km of 243 points worldwide were selected if there were data from several stations from at least a cruise in category 1 and at least, respectively a cruise from category 2. In the Pacific sector of the Southern Ocean, to cover the regions not covered by category 1 and 2 cruises, categories 3-7 were used to construct crossovers. As an example, Figure 5 shows the locations of the subset of P03 and P14 stations used for a crossover comparison for a crossover at 24.2°N and 179°E. This crossover is assigned the designation CR081E, and 
[revised manuscript text omitted]